# Endoplasmic reticulum acetyltransferases Atase1 and Atase2 differentially regulate reticulophagy, macroautophagy and cellular acetyl-CoA metabolism

Michael J. Rigby[1,2,3], Alexis J. Lawton [4], Gulpreet Kaur[5], Varuna C. Banduseela[1,7], William E. Kamm[1,3], Aparna Lakkaraju[5,8], John M. Denu[4] & Luigi Puglielli [1,2,3,6✉]

N$^\varepsilon$-lysine acetylation in the ER lumen is a recently discovered quality control mechanism that ensures proteostasis within the secretory pathway. The acetyltransferase reaction is carried out by two type-II membrane proteins, ATase1/NAT8B and ATase2/NAT8. Prior studies have shown that reducing ER acetylation can induce reticulophagy, increase ER turnover, and alleviate proteotoxic states. Here, we report the generation of Atase1$^{-/-}$ and Atase2$^{-/-}$ mice and show that these two ER-based acetyltransferases play different roles in the regulation of reticulophagy and macroautophagy. Importantly, knockout of Atase1 alone results in activation of reticulophagy and rescue of the proteotoxic state associated with Alzheimer's disease. Furthermore, loss of Atase1 or Atase2 results in widespread adaptive changes in the cell acetylome and acetyl-CoA metabolism. Overall, our study supports a divergent role of Atase1 and Atase2 in cellular biology, emphasizing ATase1 as a valid translational target for diseases characterized by toxic protein aggregation in the secretory pathway.

[1] Department of Medicine, University of Wisconsin-Madison, Madison, WI, USA. [2] Neuroscience Training Program, University of Wisconsin-Madison, Madison, WI, USA. [3] Waisman Center, University of Wisconsin-Madison, Madison, WI, USA. [4] Department of Biomolecular Chemistry and the Wisconsin Institute for Discovery, University of Wisconsin-Madison, Madison, WI, USA. [5] Department of Ophthalmology and Visual Sciences, University of Wisconsin-Madison, Madison, WI, USA. [6] Geriatric Research Education Clinical Center, Veterans Affairs Medical Center, Madison, WI, USA. [7] Present address: Department of Pharmacology, University of Michigan, Ann Arbor, MI, USA. [8] Present address: Department of Ophthalmology and Anatomy, University of California, San Francisco, CA, USA. ✉email: lp1@medicine.wisc.edu

Nε-lysine acetylation of nascent proteins within the lumen of the ER is a recently discovered quality control (QC) mechanism that ensures protein homeostasis (proteostasis) within the secretory pathway[1–18]. ER-based Nε-lysine acetylation is catalyzed by two ER-membrane bound acetyltransferases, ATase1/NAT8B and ATase2/NAT8[9]. Both ATases are type-II membrane proteins with the catalytic domain facing the ER lumen. They work in tandem with AT-1/SLC33A1, an ER-membrane antiporter that moves acetyl-CoA from the cytosol into the lumen of the ER in exchange for free CoA[8]. Dysfunctional ER acetylation, as caused by loss-of-function mutations or gene duplication events, is associated with severe inherited diseases[6,19–24].

The proteostatic functions of the ER acetylation machinery involve acetylation of correctly folded glycoproteins to ensure engagement of the secretory pathway as well as regulation of ER-autophagy (also referred to as reticulophagy or ER-phagy) to dispose of toxic protein aggregates[3,4,8,10,11]. Mechanistically, the regulation of reticulophagy involves acetylation of ATG9A, the only integral membrane autophagy protein, on two lysine residues K359 and K363 that face the lumen of the ER[11,14]. The acetylation status of ATG9A regulates its ability to interact with the reticulophagy receptors, FAM134B and SEC62, and engage cytosolic LC3β[14,15,17,25–28]. Importantly, mice with reduced ER acetylation display excessive induction of reticulophagy while mice with increased ER acetylation display the opposite[7,15,17]. In both cases, lack of homeostatic balance causes severe disease phenotypes[7,15,17]. Therefore, fluctuations in ER-based Nε-lysine acetylation have dramatic impacts on glycoprotein flux, ER turnover, and cellular physiology.

Autophagy is an essential component of the cell degradation system that is responsible for the disposal of large protein aggregates within the cell. Malfunction of autophagy contributes to the progression of many diseases across lifespan, whereas increased levels of autophagy can be beneficial in mouse models of diseases characterized by increased accumulation of toxic protein aggregates[6,14,17,29–37].

Here, we report the generation of Atase1−/− and Atase2−/− mice and show that these two ER-based acetyltransferases play different roles in the regulation of reticulophagy and macroautophagy. We also show that knockout of Atase1 alone results in activation of reticulophagy and alleviated proteotoxicity in a mouse model of Alzheimer's disease (AD). Furthermore, loss of either Atase1 or Atase2 resulted in widespread changes in the cellular acetylome but differential changes in acetyl-CoA metabolism. Overall, our study supports partially divergent functions for Atase1 and Atase2, emphasizing ATase1 as a valid translational target for diseases characterized by toxic protein aggregation in the secretory pathway.

## Results

**Knockout of Atase2 resulted in a compensatory increase in Atase1 expression in multiple organs.** In order to elucidate the roles of Atase1 and Atase2 independently, we generated Atase1−/− and Atase2−/− mice on a C57BL/6 J background. Both knockout mice were born with Mendelian ratio and did not exhibit any apparent physical abnormalities. The gene knockouts were detectable by traditional PCR (Fig. 1a,b). To confirm the gene was indeed knocked out, we designed reverse transcription quantitative PCR (RT-qPCR) primers specific for mouse Atase1 and Atase2, and in multiple tissue types, we were able to confirm that the respective gene was no longer expressed at the mRNA level (Fig. 1c).

Once the desired genetic changes were confirmed, we performed a phenotypic assessment of our Atase1−/− and

Atase2−/− mice. First, we performed necropsy and histologic assessment of all organ and tissue types in both knockout male and female mice but did not observe any notable gross or histologic abnormalities. Next, we assessed the Atase1−/− and Atase2−/− mouse behavior compared to their WT littermates by several paradigms including open field, light/dark box exploration, novel object recognition, marble burying, and fear conditioning. We did not observe substantial changes in behavior in either Atase1−/− or Atase2−/− mice (Supplementary Fig. 1). There are reported GWAS associations between ATase2/NAT8 and chronic kidney disease in humans[38–41]. Furthermore, one study put forward the hypothesis that ATase2/NAT8 might be responsible for mercapturic acid synthesis and excretion of xenobiotics in the urine[42]. Thus, we assessed kidney health in our knockout mice by specifically evaluating kidney weight, plasma creatinine and urea nitrogen, which were all not different from WT littermates (Supplementary Fig. 2a–c). We attempted to measure the spot urine albumin to creatinine ratio, but in most urine samples, the albumin level was below the limit of detection and thus could not be reliably quantified. We also measured spot urine 1,4-dihydroxynonane mercapturic acid (DHN-MA), a mercapturic acid derivative naturally excreted by the kidney, and again found no difference from WT littermates (Supplementary Fig. 2d). Additional studies have shown that reduced ER-based acetylation in hypomorphic AT-1 mice results in chronic pancreatitis and fibrosis, presumably from disrupted excretion of pancreatic enzymes[43]. In both our knockout mice, we did not observe any changes in the histologic appearance of the pancreas nor changes suggestive of pancreatic fibrosis (Supplementary Fig. 3). In conclusion, our Atase knockout mice appear healthy without evident behavioral abnormalities, organ dysfunction, or disease.

Using the primers specific for mouse Atase1 and Atase2 validated above, we performed RT-qPCR on several tissue types to examine the endogenous expression of the two genes. By using absolute quantification compared to a known amount of plasmid DNA containing either the mouse Atase1 or Atase2 gene, both genes were found to be ubiquitously expressed in the tissue types we examined; additionally, there was significantly more Atase2 expression observed in the kidney (Fig. 1d, e). By using relative quantification, we examined the expression of the Atase enzymes as well as the ER acetyl-CoA transporter At-1 to assess for potential compensation when one of the ER acetyltransferases was knocked out. In multiple tissue types, we observed an increase in Atase1 expression in the Atase2−/− mouse; there were no substantial changes in Atase2 or At-1 expression in either knockout mouse (Fig. 1f–h). The compensatory upregulation of Atase1 in the Atase2−/− mouse may play a role in the phenotype observed below.

**Knockout of Atase1 activated reticulophagy in the mouse.** Mice with reduced ER acetylation, as caused by hypomorphic AT-1, display excessive induction of reticulophagy while mice with increased ER acetylation, as caused by AT-1 overexpression, display reduced reticulophagy[7,15,17]. Therefore, we predicted that knockout of the individual acetyltransferase enzymes in the ER would result in reduced ER-based acetylation of Atg9a and activation of reticulophagy. In the following reticulophagy-focused studies, we used both liver and mouse embryonic fibroblasts (MEFs). While the liver is more congenial for the isolation and enrichment of the ER, MEFs can be more easily imaged with autophagy probes and controlled for metabolic variables.

To start, we prepared enriched liver ER from the knockout mice and examined the protein expression level of several reticulophagy-related proteins. We observed a decrease in expression of the reticulophagy receptor Fam134b and associated autophagy protein Atg9a in the Atase1−/− mouse, consistent with

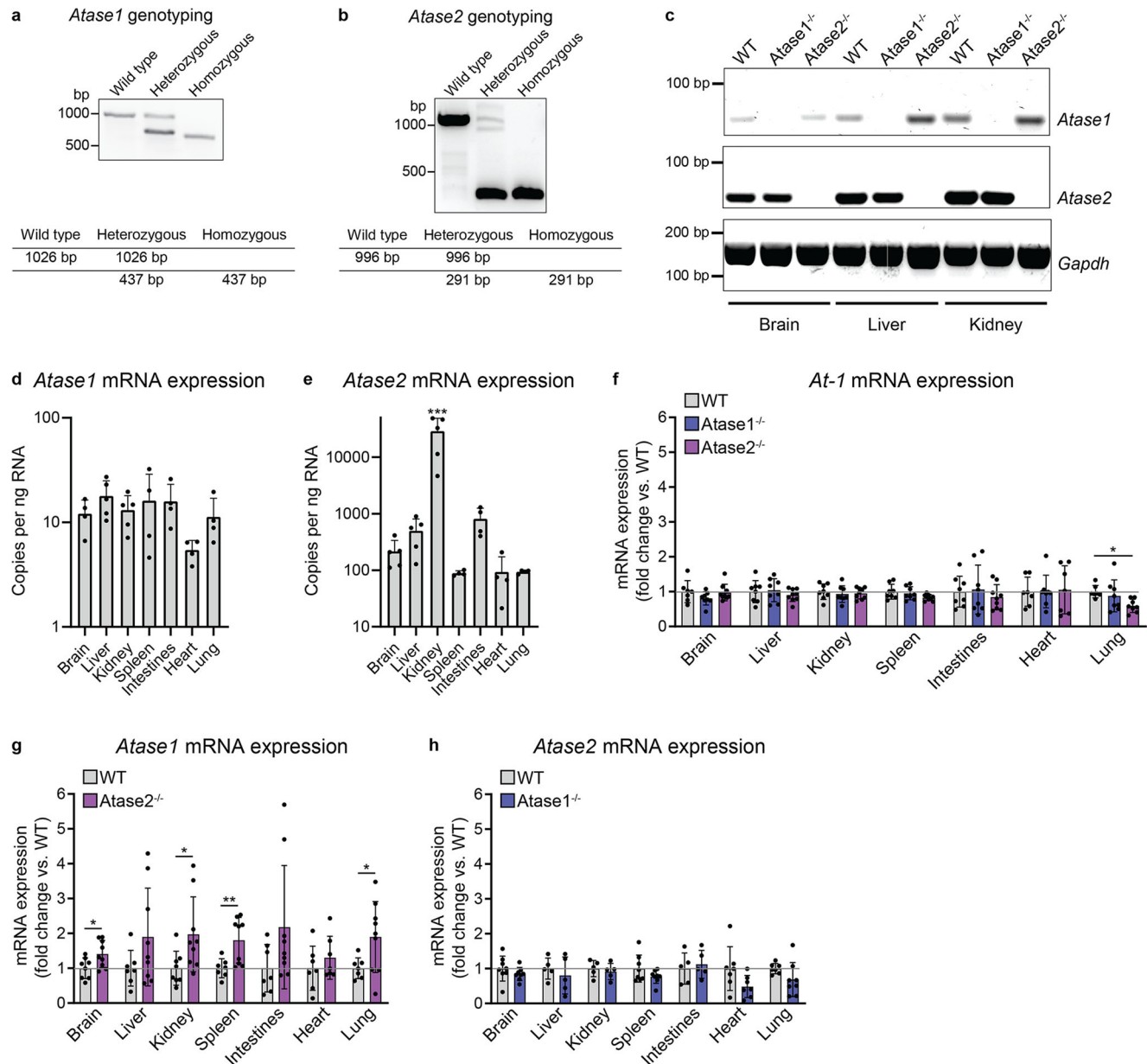

**Fig. 1 Atase1 expression is increased in the Atase2⁻/⁻ mouse. a**, **b** Representative Atase1⁻/⁻ and Atase2⁻/⁻ genotyping results. PCR bands observed when genotyping Atase1 and Atase2 knockout mice are shown with expected fragment sizes displayed in the adjacent tables. **c** RT-qPCR fragments from WT, Atase1⁻/⁻, and Atase2⁻/⁻ mouse cDNA using primers specific for mouse Atase1 (Nat8b), Atase2 (Nat8), or Gapdh. **d**, **e** Absolute RT-qPCR quantification in WT mice using Atase-specific primers. Tissue-specific expression was compared to a DNA standard curve using Nat8b-pCMV6 or Nat8-pCMV6 plasmid. Data are mean ± SD, $n = 4$–5 per genotype. ***$p < 0.0005$ via ordinary one-way ANOVA with Tukey's multiple comparison test (kidney vs. all other tissue types). **f**–**h** Relative RT-qPCR using the Atase-specific primers. Data are mean ± SD, $n = 5$–9 per genotype. *$p < 0.05$, **$p < 0.005$, ***$p < 0.0005$ via ordinary one-way ANOVA with Dunnett's multiple comparison test (At-1 expression) or Student's t test (Atase1 and Atase2 expression). All mice were male and 3 months of age at time of study.

increased turnover of the ER (Fig. 2a). To verify this change, we performed immunocytochemistry on MEFs obtained from our knockout mice and observed a decrease in Fam134b puncta in the Atase1⁻/⁻ MEFs (Fig. 2b). In addition, we observed an increase in Lc3β puncta in both knockout mice, which is consistent with activation of reticulophagy (Fig. 2b). Importantly, the change in Fam134b protein expression did not appear to be driven by a decrease in mRNA expression (Fig. 2c).

Next, we examined the acetylation status of Atg9a as well as Atg9a-interacting proteins by studying liver ER. Prior studies have shown that inhibition of the Atases results in decreased acetylation of Atg9a, increased interaction between Atg9a and

reticulophagy receptors Fam134b and Sec62, and activation of reticulophagy[14,17]. By performing an immunoprecipitation with an antibody specific for acetylated lysine residues, we found ER-based Atg9a to be hypoacetylated in Atase1⁻/⁻ mice, when compared to WT (Fig. 2d). This finding was paralleled by increased Atg9a-Fam134b and Atg9a-Sec62 interaction on the ER in Atase1⁻/⁻ mice (Fig. 2e, f). Interestingly, we did not observe changes in Atg9a acetylation status nor interaction with Fam134b/Sec62 in the Atase2⁻/⁻ mice (Fig. 2d–f).

Finally, we employed ER-specific probes to track reticulophagy and ER turnover in our knockout MEFs. First, we transfected our MEFs with the ER tandem reporter mCherry-GFP-RAMP4[44],

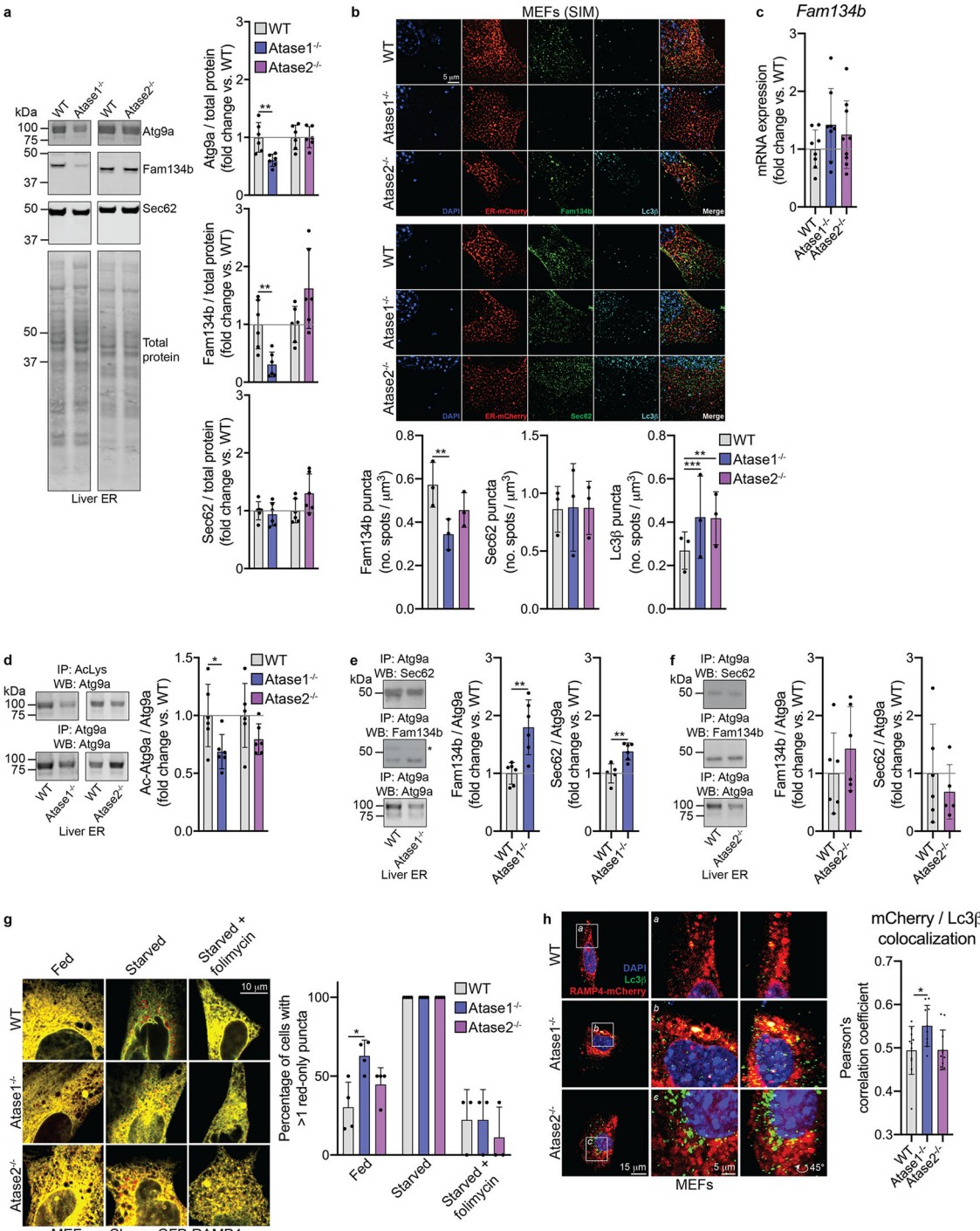

which gives a yellow signal at a neutral pH and red signal in an acidic pH due to quenching of the GFP signal. By quantifying the number of red puncta per cell, which represents acidified ER from autophagolysosome formation, we observed an increase in the percentage of cells that contained more than one ER-containing autophagolysosome in the Atase1$^{-/-}$ MEFs compared to WT MEFs (Fig. 2g). Importantly, this percentage increased as expected when the MEFs were starved in Earl's balanced salt solution (EBSS) and decreased to fed levels when EBSS was supplemented with folimycin, which prevents lysosomal acidification (Fig. 2g), thus providing functional validation to our assay. Next, we transfected our MEFs with RAMP4-mCherry to specifically label the ER followed by immunostaining for Lc3β.

We observed an increase in co-localization between the RAMP4-mCherry and Lc3β signals in the Atase1$^{-/-}$ MEFs compared to WT MEFs, suggesting increased autophagosome formation from the ER (Fig. 2h). As expected, we did not observe changes between WT and Atase2$^{-/-}$ MEFs in either of these experiments.

When taken together, the above results show that knockout of Atase1 but not Atase2 in the mouse causes elevated reticulophagy and ER turnover.

**Atase1$^{-/-}$ and Atase2$^{-/-}$ mice exhibited increased macro-autophagy and ER stress.** In addition to activation of reticulophagy, prior studies have demonstrated that reduced ER-based acetylation also causes an activation of general cellular autophagy

**Fig. 2 Atase1$^{-/-}$ mice display activated reticulophagy. a** Atg9a, Fam134b, and Sec62 Western blotting using enriched liver ER. Levels of Atg9a, Fam134b, or Sec62 are normalized to total protein. Data are mean ± SD, $n = 6$ animals per genotype. **$p < 0.005$ via Student's $t$ test. **b** Mouse embryonic fibroblasts (MEFs) transiently transfected with ER-mCherry plasmid and immunostained for Fam134b, Sec62, and Lc3β. Representative structured illumination microscopy images are displayed. Data are mean ± SD, $n = 3$ embryo lines per genotype. **$p < 0.005$, ***$p < 0.0005$ via ordinary one-way ANOVA with Dunnett's multiple comparison test. **c** mRNA expression of *Fam134b* in liver normalized to *Gapdh* expression. Data are mean ± SD, $n = 8$–9 animals per genotype. **d** Immunoprecipitation against acetylated lysine (AcK) or Atg9a on enriched liver ER solubilized with 1% Triton X-100. Levels of acetylated Atg9a (Atg9a-Ac) was normalized to total Atg9a. Data are mean ± SD, $n = 5$–6 per genotype. *$p < 0.005$ via Student's $t$ test. **e, f** Co-immunoprecipitation against Atg9a and Western blotting against either Sec62, Fam134b, or Atg9a using enriched liver ER. *Denotes a non-specific band. Data are mean ± SD, $n = 5$–6 per genotype. **$p < 0.005$ via Student's $t$ test. All mice were male and 3 months of age at time of study. **g** Live-cell imaging of MEFs transiently transfected with the mCherry-GFP-RAMP4 tandem reporter plasmid. Representative single z-plane confocal images are displayed. Red-only puncta were counted for each cell imaged and percentage of cells in a given embryo line with more than one red-only puncta are reported. Data are mean ± SD with each dot representing an embryo line, $n = 3$–4 embryo lines per genotype. *$p < 0.05$ via ordinary one-way ANOVA with Tukey's multiple comparison test; results are only shown within nutrient conditions. **h** MEFs transiently transfected with the mCherry-RAMP4 plasmid and immunostained for Lc3β. Representative confocal images are displayed. Data are mean ± SD with each dot representing a single cell with $n = 3$ embryo lines per genotype. *$p < 0.05$ via ordinary one-way ANOVA with Dunnett's multiple comparison test.

(also referred to as macroautophagy) as well as ER stress[14,15]. Therefore, we investigated whether this also occurred in our Atase knockout mice. First, we performed Western blotting of several commonly used markers of autophagic flux in our knockout MEFs. We observed an increase in expression of Lc3β-I, Lc3β-II, and Beclin consistent with chronic activation of macro-autophagy in the Atase1$^{-/-}$ and Atase2$^{-/-}$ MEFs compared to WT (Fig. 3a). In liver, we also observed increased expression of Lc3β-I in both knockout mice as well as an increase in Beclin in Atase1$^{-/-}$ mice (Supplementary Fig. 4). We did not observe a change in p62 expression in our MEFs, but in liver, we did observe a decrease in expression levels in both knockout mice compared to WT supportive of increased autophagic flux (Fig. 3b). Finally, we performed live cell imaging in our MEFs with GFP-LC3β, which represents autophagosome trafficking throughout the cell (Fig. 3c). Overall, we observed an increase in organelle density in both knockout MEFs compared to WT consistent with prior data (see Figs. 2b, 2h). In addition, in our Atase1$^{-/-}$ MEFs, we observed an increase in autophagosome speed and track displacement length compared to WT MEFs (Fig. 3c). Thus, these data show that both knockout mice have an increase in autophagic flux, with a more dramatic phenotype observed in the Atase1$^{-/-}$ mice.

Next, we evaluated levels of ER stress in our Atase knockout mice. By Western blotting from liver lysates, we observed an increase in BiP/Grp-78 expression in both the Atase1$^{-/-}$ and Atase2$^{-/-}$ mice compared to WT (Fig. 3d). While not statistically significant, we did observe a trend in increased BiP mRNA expression in the liver of both knockouts compared to WT mice, with large variation in the Atase2$^{-/-}$ mice (Fig. 3f). We then evaluated for activation of the three canonical ER stress signaling pathways to assess for drivers of increased BiP expression. Both the Perk and Ire1 pathways did not appear activated as evident by the lack of change in the phosphorylation status of Perk, eIF2α, and Ire1; nuclear expression level of Atf4; and mRNA expression of spliced *Xbp1* and *Atf4* (Fig. 3d–f). However, we did observe an increase in the cleavage of Atf6 in the Atase2$^{-/-}$ mouse as evident by increased expression of the p50 fragment relative to the p90, full length form of the protein (Fig. 3d). In addition, by transfecting MEFs with an Atf6 transcriptional reporter that results in the expression of firefly luciferase, we observed an increase in luminescence in our Atase2$^{-/-}$ MEFs, consistent with an increase in p50-Atf6 activity (Fig. 3g). Overall, our results show activation of the Atf6 canonical ER stress signaling pathway in the Atase2$^{-/-}$ mouse together with stimulation of macroautophagy.

**Knockout of Atase1 improves the proteotoxicity phenotype of the APP/PS1 mouse model of Alzheimer's disease.** Previous studies have shown that induction of reticulophagy, down-stream of

the ER acetylation machinery in AT-1 hypomorphic (AT-1$^{S113R/+}$) mice or in the presence of ATase1/ATase2 chemical inhibitors, can resolve the proteopathy associated with the AD phenotype[14]. Since our Atase knockout mice exhibit activation of reticulophagy (Atase1$^{-/-}$ only) and macroautophagy (both Atase1$^{-/-}$ and Atase2$^{-/-}$), we crossed our Atase knockout mice with the APP$_{swe}$/PS1dE9 (henceforth referred to as APP/PS1) AD-like model to generate APP/PS1;Atase1$^{-/-}$ and APP/PS1;Atase2$^{-/-}$ mice. First, we tracked lifespan over 10 months, which was dramatically reduced in the APP/PS1 mouse. We observed increased survival in both APP/PS1;Atase1$^{-/-}$ and APP/PS1;Atase2$^{-/-}$ mice compared to APP/PS1 mice; however, the rescue of survival was more evident in the APP/PS1;Atase1$^{-/-}$ mouse (Fig. 4a). In addition, while male mice lifespan was partially rescued in both crosses, we did not observe a rescue in lifespan for the female APP/PS1;Atase2$^{-/-}$ mice compared to APP/PS1 female mice (Fig. 4a).

At 10 months of age, we assessed the male APP/PS1 phenotype compared to the APP/PS1;Atase1$^{-/-}$ and APP/PS1;Atase2$^{-/-}$ male mice, evaluating several pathologic hallmarks of AD including amyloid plaque deposition, gliosis, and synaptic loss. First, by using thioflavin-S to stain for dense plaques in the brain, we observed a reduction in plaque density and area coverage in our APP/PS1;Atase1$^{-/-}$, most notable in the hippocampus; whereas we only observed a slight reduction in plaque area percentage in the APP/PS1;Atase2$^{-/-}$ mice (Fig. 4b). Next, we assessed astrocytic and microglial activation via immunofluorescence and Western blotting in the cortex. In our APP/PS1 mice, we observed a marked increase in number of astrocytes and microglia via Gfap and Iba1 immunofluorescence, respectively, compared to WT control mice (Fig. 4c). This change in immunostaining was also evident via Western blotting for Gfap and Iba1 in which the expression of these proteins was significantly increased in the APP/PS1 mice compared to WT (Fig. 4d). In our APP/PS1;Atase1$^{-/-}$ and APP/PS1;Atase2$^{-/-}$ mice, we observed a significant decrease in Gfap immunofluorescence signal as well as expression level while the Iba1 immunofluorescence signal and expression level were mostly unchanged (Fig. 4c, d). Finally, we evaluated for synaptic loss in the CA3 region of the hippocampus via immunofluorescent staining for the presynaptic protein synaptophysin and the postsynaptic protein Psd-95. In our APP/PS1 mice, we observed reduced synaptophysin immunofluorescent signal in the CA3 region as well as a marked reduction in the number of synaptophysin and Psd-95 co-localized puncta compared to WT mice (Fig. 4e). In both our APP/PS1;Atase1$^{-/-}$ and APP/PS1;Atase2$^{-/-}$ mice, there was an increase in the number of co-localized puncta compared to APP/PS1 mice, signifying more synapses retained at 10 months of age (Fig. 4e). Overall, our data show that knockout of either Atase1 or Atase2 in the mouse can

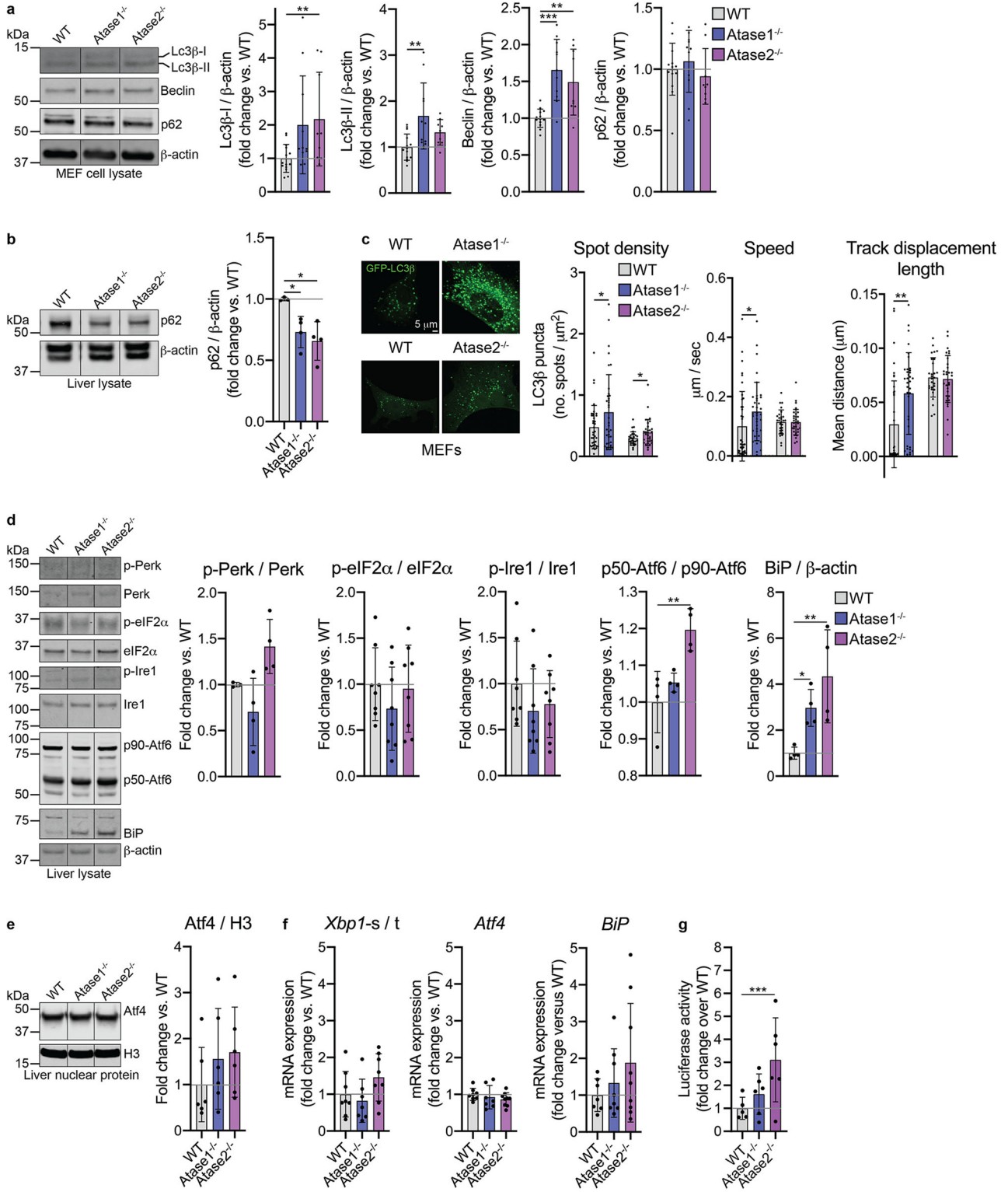

rescue features of the APP/PS1 mouse AD-like phenotype, namely lifespan, amyloid plaque deposition, gliosis, and synapse loss, with knockout of Atase1 providing a more robust rescue.

**Lysine acetylation stoichiometry revealed widespread changes reflecting overlapping and non-overlapping substrates of Atase1 and Atase2 in the secretory pathway.** Both Atase enzymes are responsible for the acetylation of ER-resident and cargo proteins; however, it is currently unknown whether they

have different substrates or whether they compete for the same substrates, either at the level of the protein or the specific lysine residue. To address both questions, we prepared enriched ER from the liver of WT, Atase1$^{-/-}$, and Atase2$^{-/-}$ mice and used a mass spectrometry method that reveals the stoichiometry of lysine acetylation within the proteome[2,45]. We detected a total of 6242 acetylpeptides, with 253 and 219 acetylpeptides in the Atase1$^{-/-}$ and Atase2$^{-/-}$, respectively, that exhibited a stoichiometry value that was statistically different from WT.

**Fig. 3 Atase1$^{-/-}$ and Atase2$^{-/-}$ mice exhibit activated macroautophagy but only Atase2$^{-/-}$ triggers the Atf6 canonical ER stress pathway. a**
Western blotting of MEF cell lysates for autophagy markers Lc3β, Beclin, and p62. Representative images are shown. Data are mean ± SD, $n = 9$–14 embryo lines per genotype. **$p < 0.005$, ***$p < 0.0005$ via ordinary one-way ANOVA with Dunnett's multiple comparison test. **b** Western blotting of liver lysate for p62. Representative images are shown. Data are mean ± SD, $n = 3$–4 mice per genotype. *$p < 0.05$ via ordinary one-way ANOVA with Dunnett's multiple comparison test. **c** MEFs were transduced (for WT/Atase1$^{-/-}$) or transfected by nucleofection (WT/Atase2$^{-/-}$) with LC3β-GFP and live imaging performed 20–24 h post transduction/transfection. Data as mean ± SD with each dot representing a single cell with $n = 3$ embryo lines per genotype. *$p < 0.05$, **$p < 0.005$ via Student's $t$ test. **d** Western blotting of total liver lysate. Data are mean ± SD, $n = 3$–8 animals per genotype. *$p < 0.05$, **$p < 0.005$ via ordinary one-way ANOVA with Dunnett's multiple comparison test. **e** Western blotting of liver nuclear protein. Data are mean ± SD, $n = 6$ animals per genotype. **f** RT-qPCR of liver cDNA. Expression of target genes were normalized to *Gapdh*. Data are mean ± SD, $n = 8$ animals per genotype. **g** Luciferase assay in mouse embryonic fibroblasts. The p5xAtf6-GL3 plasmid was co-transfected with the pNL1.1 PGK [Nluc/PGK] internal control plasmid (to normalize for transfection efficiency) by nucleofection, and both luciferase and nanoluciferase activity was measured 24 h post-transfection. Data are expressed as the mean normalized ratio of luciferase to nanoluciferase activity (±SD). Each data point represents an independent embryo cell line, $n = 5$–6 per genotype. ***$p < 0.0005$ via ordinary two-way ANOVA (genotype vs. embryo line) with Dunnett's multiple comparison test. Representative Western blot images contain non-adjacent lanes on the same membrane separated by vertical lines. All mice are male and 3 months old at time of study.

We first evaluated acetylpeptides found within the secretory pathway as possible direct substrates of Atase1 and Atase2. Of the 6242 acetylpeptides detected, we filtered these based upon a cellular localization of "ER", "Golgi", or "secreted", yielding a total of 2396 acetylpeptides. Of these acetylpeptides, we found 88 and 82 sites in the Atase1$^{-/-}$ and Atase2$^{-/-}$, respectively, that exhibited a stoichiometry value that was statistically different from WT (Fig. 5a). Interestingly, when evaluating these acetylpeptide stoichiometry changes from WT between the Atase1$^{-/-}$ and Atase2$^{-/-}$ mice, the overall distributions of the two were different. The Atase2$^{-/-}$ distribution centered below zero, suggesting a net decrease in lysine acetylation in the secretory pathway (Fig. 5b). When comparing the acetylpeptides that significantly changed from WT, only 19 sites overlapped between the two knockout mice, but all except 3 exhibited a stoichiometry change from WT in the same direction (Fig. 5c). In addition, when examining the non-overlapping sites, there was a significant positive correlation between the Atase1$^{-/-}$ and Atase2$^{-/-}$ changes in acetyl stoichiometry (Supplementary Fig. 5). Next, we examined the acetylpeptides that could be detected in the WT samples with confidence ($n \geq 3$) but not in the Atase1$^{-/-}$ or Atase2$^{-/-}$ samples; we found 7 and 2 sites in the Atase1$^{-/-}$ and Atase2$^{-/-}$, respectively, that we call "Atase1- or Atase2-dependent" (Fig. 5d). These acetylpeptides do not appear in the volcano plots shown in Fig. 5a since a stoichiometry difference cannot be computed. Importantly, we were able to detect the heavy-labeled acetylpeptides (in vitro acetylation) in the knockout mice, but not the light-labeled acetylpeptides (endogenous acetylation), verifying a stoichiometry value of effectively zero.

To evaluate the biologic significance of the above findings, we combined the sites with significantly different stoichiometry (Fig. 5a) with those that appeared Atase-dependent (Fig. 5d) and performed a gene ontology analysis of the proteins harboring those sites (Fig. 5e). In both knockouts, we found many overlapping cellular component categories such as ER, ER-Golgi-Intermediate Compartment, and Golgi-associated, which highlight known functions of the Atases within the secretory pathway. However, Atase1$^{-/-}$ mice displayed a preponderance of categories that are generally involved with the engagement as well as morphology of the secretory pathway, likely suggesting partially different regulatory functions of the two transferases (Fig. 5e). Interestingly, we also found ribosome-associated categories, which would suggest a previously uncharacterized role of the Atases in post-translational modification of ribosome-associated proteins. The representation of ribosomal-associated proteins within our enriched ER is not surprising as the great majority of ribosomal elements are tightly associated with the ER membrane and about 50–75% of the translational activity of the ER is directed toward cytosolic proteins[46].

N$^\varepsilon$-lysine acetyltransferases rely upon the tertiary structure of the substrate protein as there is no known consensus motif for N$^\varepsilon$-lysine acetylation[6]. As such, changes in protein folding are expected to alter exposure and/or recognition of potential lysine sites. Furthermore, changes in the acetylation of one site might affect the acetylation of a different site that resides close within the tertiary structure of the protein. Indeed, we observed 11 sites that were not detected in the WT samples but were detected in either Atase knockout with confidence ($n \geq 3$ in either Atase knockout and heavy-labeled acetylpeptide detected in WT). These "new sites" likely reflect changes in protein folding or simply competition between structurally close lysine sites (Supplementary Fig. 6). When taken together, the above data indicate that lack of one Atase results in complex adaptive changes across the entire acetylome of the secretory pathway, which might reflect their overlapping abut also different biological functions.

**Atase1$^{-/-}$ and Atase2$^{-/-}$ affected non-secretory pathway protein acetylation and acetyl-CoA metabolism.** Since our acetylomic data revealed many changes in protein acetylation within the secretory pathway, we next decided to examine whether lack of one Atase in our knockout mice would also affect acetyl-CoA/CoA dynamics and cause adaptive responses beyond the ER. In fact, prior studies in our laboratory have shown that changes in At-1 activity, the ER membrane-based acetyl-CoA/CoA antiporter, also affect cellular acetylation and lipid metabolism beyond the ER[2]. We reasoned that reduced acetyl-CoA consumption within the ER lumen, as caused by the disruption of one of the acetyltransferases, would limit the generation and availability of free CoA, which is necessary for the antiporter-mediated influx of acetyl-CoA from the cytosol (Fig. 6a).

As mentioned above, when we decided to determine stoichiometric changes within the acetylome of Atase1$^{-/-}$ and Atase2$^{-/-}$ mice, we opted to use enriched- instead of purified-ER because, in addition to abundant ER membranes, the former retains sufficient residual cytosolic material that would allow us to explore possible perturbation of the acetyl-CoA/CoA antiporter system, as reflected by the "non-secretory pathway acetylome". Indeed, our strategy successfully yielded 3846 acetylpeptides not designated with a cellular localization of "ER", "Golgi", or "secreted" (henceforth called "non-secretory pathway"). Of these 3846 acetylpeptides, we found 165 and 134 sites in Atase1$^{-/-}$ and Atase2$^{-/-}$ mice, respectively, that exhibited a stoichiometry value that was statistically different from WT (Fig. 6b). When evaluating these acetylpeptide stoichiometry changes from WT between Atase1$^{-/-}$ and Atase2$^{-/-}$ mice, the Atase1$^{-/-}$ distribution centered well above zero, suggesting a net increase in lysine acetylation (Fig. 6c). When comparing the acetylpeptides that significantly changed from WT, 31 sites overlapped between the

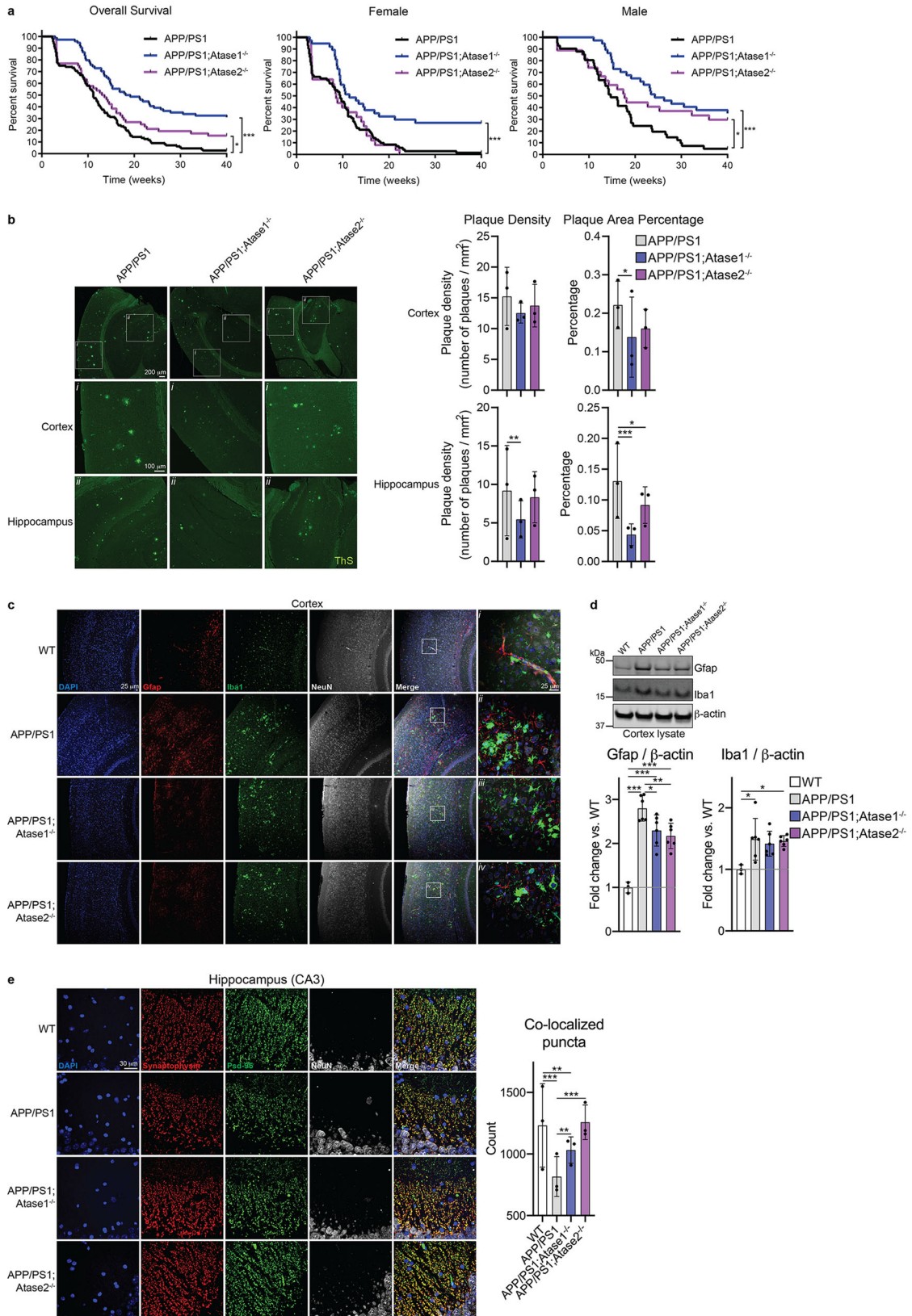

two knockout mice, and all except one site exhibited the same direction of stoichiometry change from WT (Fig. 6d). Again, the changes in acetyl stoichiometry from WT of non-overlapping sites in both mouse models displayed a positive correlation (Supplementary Fig. 5). When evaluating these significantly changed acetylpeptides with a gene ontology analysis on the proteins

harboring the sites, we found several categories in both knockouts that highlight a broad cellular response (Fig. 6e). Interestingly, the Atase2$^{-/-}$ analysis revealed a heavy predominance of proteasome categories that did not appear in the Atase1$^{-/-}$ analysis.

We then examined the steady-state levels of acetyl-CoA and CoA in the liver cytosol and ER in fasted mice. We found a

**Fig. 4 Atase1$^{-/-}$ and Atase2$^{-/-}$ improve the APP/PS1 proteotoxicity phenotype. a** Combined (both male and female) and sex-specific survival analysis. Data represented are as follows: APP/PS1: $n = 112$, 41 males, 71 females; APP/PS1; Atase1$^{-/-}$: $n = 74$, 37 males, 37 females; APP/PS1; Atase2$^{-/-}$: $n = 52$, 27 males, 25 females. *$p < 0.05$, ***$p < 0.0005$ via log-rank test. **b** Thioflavin-S staining of 5 μm paraffin-embedded brain slices. Data are mean ± SD with each data point being the average of technical replicates (a single brain slice) for a given animal, $n = 3$ animals per genotype. *$p < 0.05$, **$p < 0.005$, ***$p < 0.0005$ via ordinary two-way ANOVA (genotype vs. animal) with Dunnett's multiple comparison test. **c** Immunostaining of 5 μm paraffin-embedded brain slices for astrocytes (Gfap) and microglia (Iba1). **d** Western blotting for Gfap and Iba1 with representative images shown. Data are normalized to β-actin and represented as the fold change over WT, mean ± SD, $n = 3$–6 mice per genotype. *$p < 0.05$ via ordinary one-way ANOVA with Tukey's multiple comparison test. **e** Immunostaining of 5 μm paraffin-embedded brain slices for the presynaptic marker synaptophysin and postsynaptic marker Psd-95. Data are mean ± SD with each data point being the average of technical replicates (a single brain slice) for a given animal, $n = 3$ animals per genotype. **$p < 0.005$, ***$p < 0.0005$ via ordinary two-way ANOVA (genotype vs. animal) with Tukey's multiple comparison test. All analysis was conducted on male mice at 10 months of age.

---

decrease in ER CoA levels in both Atase knockout mice (Fig. 6f). Since cytosolic acetyl-CoA is the substrate for fatty acid synthesis, we assessed whether our Atase knockout mice had an accumulation of lipids in the liver as seen when At-1 activity is reduced[2]. While liver H&E staining did not reveal dramatic changes in liver microscopic anatomy, lipid-specific staining did reveal an increase in lipid droplet density in the Atase2$^{-/-}$ mouse (Fig. 6g, h).

When taken together, the above data support the argument that hyperacetylation of cytosolic proteins in Atase1$^{-/-}$ as well as the accumulation of lipid droplets in Atase2$^{-/-}$ reflect secondary adaptation that may be from disruption of At-1 acetyl-CoA/CoA antiporter activity. They also suggest that the Atases might be part of an intricate system that regulates metabolic crosstalk between the ER lumen and the cytosol, presumably through At-1.

## Discussion

N$^\varepsilon$-lysine acetylation within the lumen of the ER, which is carried out by the two acetyltransferases ATase1/NAT8B and ATase2/NAT8, plays an important role in the regulation of proteostasis within the organelle[6]. There is building evidence that ATase1 and ATase2 might serve different roles in cellular biology. Indeed, despite high sequence similarity and catalytic activity, recent work has shown that ATase1 is post-translationally regulated by acetylation while ATase2 is transcriptionally regulated by the immediate-early gene cascade[3,4,18]. The work presented here suggests that Atase1 plays a more substantial role in the regulation of reticulophagy and proteostasis, as also revealed by the rescue of several phenotypic qualities in the APP/PS1 mouse that may in part be due to the acetylation status of Atg9a and downstream assembly of the reticulophagy machinery. We presume that Atase1$^{-/-}$ mice exhibit increased turnover of toxic protein aggregates within the secretory pathway that ameliorates the APP/Aβ-associated proteotoxicity in the APP/PS1 mouse. A similar effect was observed with APP$_{695/swe}$ mice, a different model of AD, following genetic or biochemical inhibition of ER acetylation[14]. It is important to point out that the lack of activation of reticulophagy in the Atase2$^{-/-}$ mouse may be due -at least in part- to the compensatory upregulation of Atase1 expression. These data also demonstrate the overlapping functions of Atase1 and Atase2 as knockout of a single ER-based acetyltransferase does not result in the same, dramatic phenotype as observed with the At-1$^{S113R/+}$ mouse[14,15]. Although many aspects of the APP/PS1 AD-like phenotype improved in one or both knockout mice, the rescue was not as dramatic as seen with the At-1$^{S113R/+}$ or compound 9 (Atase1/Atase2 inhibitor) treatment. However, it is imperative to point out that those prior studies targeted both Atases and used the less-severe APP$_{695/swe}$ mouse that does not harbor the PS1dE9 mutation. Overall, this work emphasizes the validity of using inhibitors of ATase1 for translational medicine for diseases characterized by toxic protein aggregation in the secretory pathway. Importantly, there are natural *ATase1* and *ATase2* nonsense mutations in humans that do not appear to associate with pathologic consequences, further emphasizing the ability to use these enzymes as therapeutic targets[6].

While Atase1 appears to be heavily implicated in the induction of reticulophagy, our data suggest that Atase2 is more involved in protein QC as loss of the enzyme results in activation of ER stress via Atf6-mediated signaling. Interestingly, Atase1 is allosterically regulated by availability of acetyl-CoA, while Atase2 is not[18]. Therefore, we could presume that Atase2 acts as a "constitutive" acetyltransferase, thus explaining the apparent activation of the ER stress response in the Atase2$^{-/-}$ mice, while Atase1 acts as "regulated" acetyltransferase. Atase2$^{-/-}$ mice also exhibited activated macroautophagy similar to the Atase1$^{-/-}$ mouse, with a mechanism appearing to be independent of the acetylation status of Atg9a and potentially reliant upon the unfolded protein response[47,48]. Activating cellular autophagy has been repeatedly shown to be beneficial in models of AD[36,49,50], which may explain the partial phenotypic rescue of the APP/PS1;Atase2$^{-/-}$ mice observed in this study. Again, it is likely that overlapping functions between Atase1 and Atase2 prevent widespread activation of ER stress and cellular pathology in the knockout mice, which is further supported by the short list of Atase1/2-dependent acetylation sites found in liver ER. Prior work in our laboratory has demonstrated that ER-based N$^\varepsilon$-lysine acetylation as catalyzed by ATase1 and ATase2 plays a role in protein QC within the secretory pathway, and alterations in levels of ER acetylation can dramatically impact glycoprotein flux[3,7,18,51]. Therefore, we have ongoing studies to assess the engagement of the secretory pathway in our knockout mice that may deepen our understanding about the differing and overlapping roles of ATase1 and ATase2.

Reduced activity of Atase1 or Atase2 are expected to have two important consequences: (1) reduced acetylation of ER-resident and -cargo proteins and (2) reduced ER acetyl-CoA consumption resulting in decreased CoA generation. AT-1, the ER-based acetyl-CoA transporter, is an antiporter that moves acetyl-CoA from the cytosol to ER in exchange for CoA from the ER to cytosol[8]. Therefore, the concentration gradients of both acetyl-CoA and CoA in both cellular compartments would impact the function of AT-1. Prior work in our lab has shown that reducing At-1 activity can dramatically alter acetyl-CoA pools within the cell, drastically changing the proteome and acetyl-proteome as well as resulting in cytosolic lipid accumulation from excess acetyl-CoA availability[2]. As a result, in our Atase knockout mice, we anticipated a similar effect due to reduced CoA generation in the ER, reduced At-1 activity, and a buildup of cytosolic acetyl-CoA. Our data demonstrated an expected decrease in steady-state ER CoA levels in both knockout mice, but acetyl-CoA levels did not change in either cellular compartment. Furthermore, Atase2$^{-/-}$ mice had evidence of cytosolic lipid accumulation whereas Atase1$^{-/-}$ did not. In addition, Atase1$^{-/-}$ mice exhibited robust hyperacetylation of non-secretory pathway proteins. Therefore, we presume that the excess cytosolic acetyl-CoA

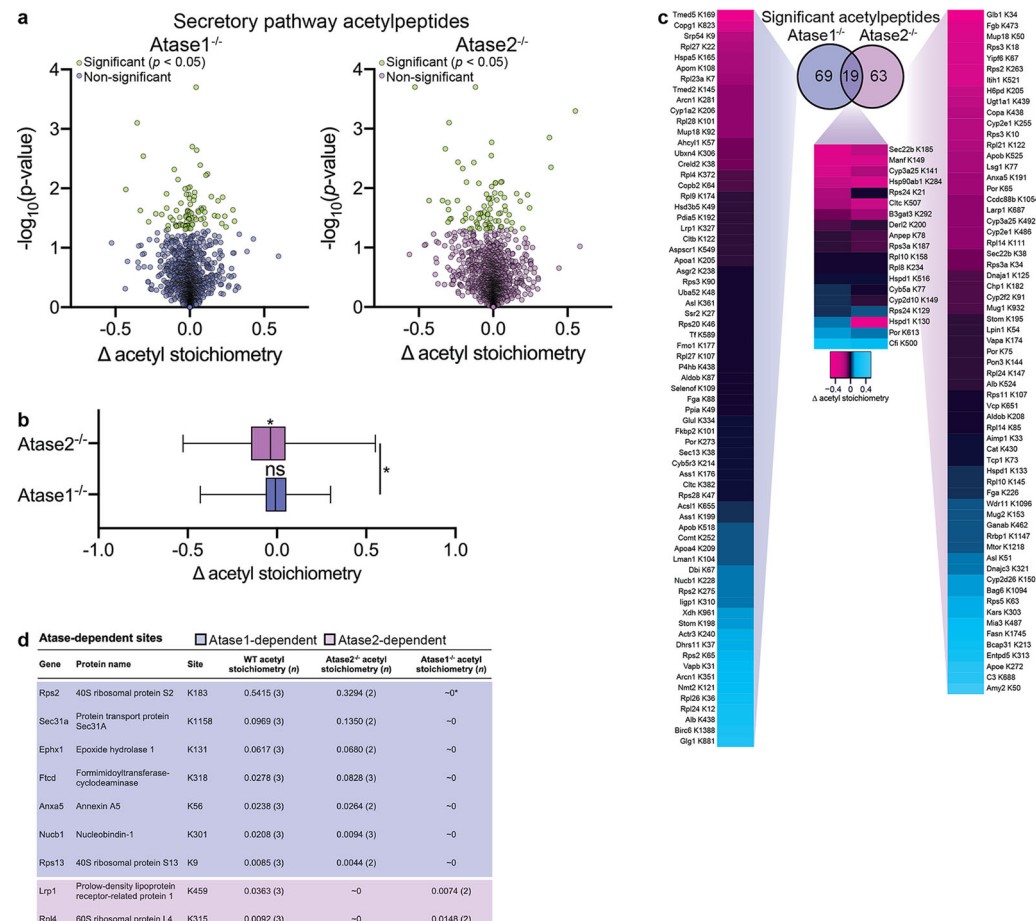

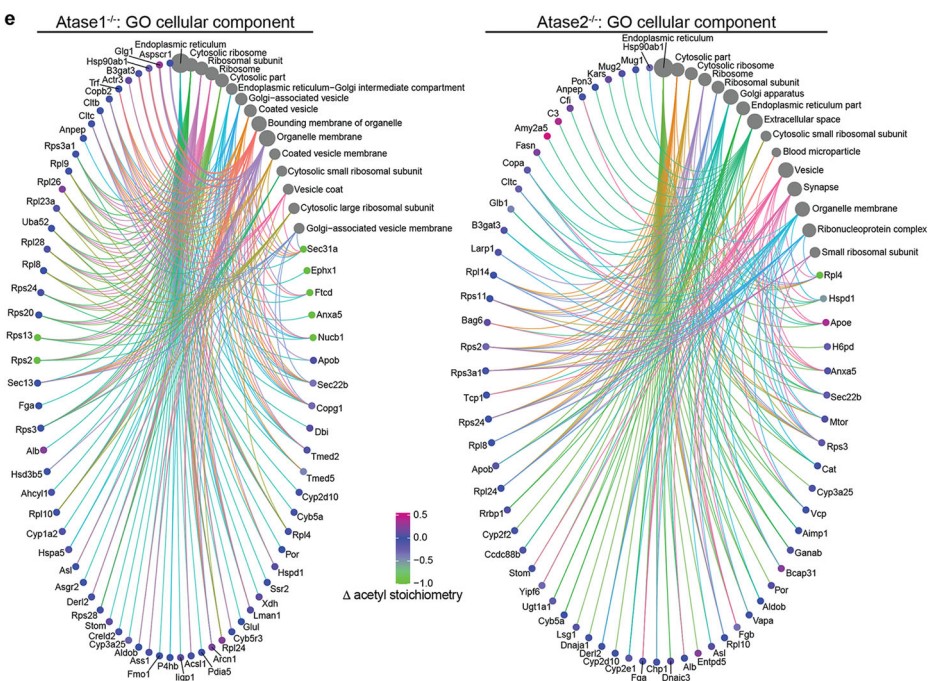

accumulating from slowdown of At-1 antiporter activity is being used to acetylate proteins in the Atase1$^{-/-}$ mouse and to build lipids in the Atase2$^{-/-}$ mouse. The underlying reasons for this different adaptive response remain to be explored.

N$^\varepsilon$-lysine acetylation is a key regulatory mechanism of many cellular proteins in eukaryotic cells. In our acetylomic data, we

observed changes in stoichiometry of acetylation both within and outside the secretory pathway with substantial overlap between Atase1$^{-/-}$ and Atase2$^{-/-}$ mice. Indeed, we only observed a handful Atase-dependent sites that are likely to only be substrates of Atase1 or Atase2, thus confirming that the vast majority of N$^\varepsilon$-lysine acetylation in the ER can be carried out by both Atase1 and

**Fig. 5 Atase1⁻/⁻ and Atase2⁻/⁻ show changes in stoichiometry of lysine acetylation within the secretory pathway. a** Volcano plots of lysine acetylation stoichiometry from enriched liver ER from WT, Atase1⁻/⁻, and Atase2⁻/⁻ mice. Only acetylpeptides that have an identified subcellular localization of "ER," Golgi," or "secreted" are included. **b** Comparison between significantly changed acetylpeptides shown in (**a**). Box plots display 25–75th percentile with a line at the mean value and whiskers at the 1st and 99th percentiles. Statistical testing between the Atase1⁻/⁻ and Atase2⁻/⁻ distributions was conducted via the Kolmogorov–Smirnov test; *$p < 0.05$. Labels over the mean value designate results of a one-sample $t$-test comparing to a mean value of 0; *$p < 0.05$; ns not significant. **c** Overlap between significantly changed acetylpeptides shown in (**a**) with a heatmap showing the stoichiometry of lysine acetylation changes from WT. **d** Acetylation sites that are detected and quantified in WT liver ER ($n \geq 3$ mice) but undetected in Atase1⁻/⁻ or Atase2⁻/⁻ liver ER. Only acetylpeptides that have an identified subcellular localization of "ER," Golgi," or "secreted" are included. *Stoichiometry value is effectively zero as the light-labeled acetylpeptide (endogenous acetylation) was not detected but the heavy-labeled acetylpeptide (in vitro acetylation) was detected. **e** Gene ontology (GO) overrepresentation analysis of the secretory pathway proteins harboring the acetylation sites that either significantly changed from WT, as shown in (**a**), and were eliminated, as shown in (**d**). Deacetylated site changes in acetyl stoichiometry was set to −1. The dot size of each network category is scaled by number of proteins included in that category. The top 15 enriched categories are shown that have been filtered by a false discovery rate threshold of $q = 0.05$.

Atase2. However, we did observe stoichiometry changes in one model that were not observed in the other suggesting that, under normal conditions, many sites are "preferentially" modified by only one Atase. These findings are likely explained by the different $K$m and affinity for the individual peptides and lysine sites. Interestingly, several of the proteins harboring unique sites exhibited changes in stoichiometry of acetylation at other sites; for example, the Nucb1 K301 site was lost in the Atase1⁻/⁻ mouse, but the K288 site exhibited increased stoichiometry in the Atase1⁻/⁻ mouse compared to WT. Furthermore, we found 11 new acetylation sites in our knockout mice that were not detected in WT samples, which may be of importance for the phenotypes observed. For example, ATL2 plays a role in remodeling the ER membrane in preparation for reticulophagy[44], and acetylation at K314 may impact the protein's function in the Atase knockout mice. Overall these data point to complex changes occurring in protein acetylation, presumably from changes in protein folding, affinity for individual sites, or acetyl-CoA availability in the knockout mice. It is important to point out that our samples were enriched for ER-resident proteins, and thus the results of our analysis are likely to miss many cargo proteins transiting through the secretory pathway. In addition, the observed changes in acetylation in our knockout mice highlight the multifaceted adaptive response of the cell imparted upon by changes in Atase1 and Atase2 activity. Indeed, in addition to the immediate regulation of ER intraluminal events, namely the engagement of the secretory pathway and the induction of reticulophagy, the ER acetylation machinery is emerging as a vital metabolic switch that ensures crosstalk among different intracellular organelles and compartments[2,6,51].

In conclusion, we have begun dissecting the differing roles of the ER-based ATase1 and ATase2 enzymes in cellular biology, demonstrating ATase1 to be a valid target for translational medicine in diseases characterized by toxic protein aggregation in the secretory pathway.

## Methods

### Atase knockout mouse generation

*Atase1⁻/⁻ mouse generation.* The mouse *Nat8b* gene was completely excised by the Genome Editing and Animal Models Core at the University of Wisconsin-Madison (Madison, WI, USA). Murine bacterial artificial chromosome (BAC) clones generated from C57BL/6 embryonic stem (ES) cell DNA and containing all of the *Nat8b* gene were purchased from Children's Hospital Oakland Research Institute (Oakland, CA). Portions of these BAC clones were used to construct a *Nat8b*-knockout (KO) targeting vector containing a floxed Neo selection cassette via traditional cloning techniques and recombineering[52]. The completed targeting vector was linearized and introduced by electroporation into murine JM8A3 (C57BL/6N-Aᵗᵐ¹ᴮʳᵈ) parental ES cells. Cells that integrated the targeting vector were selected by G418; gancyclovir (GANC) was also used to selected against clones that contain the HSV-TK cassette, thus enriching for clones that integrated the Neo cassette by homologous recombination. Colonies resistant to G418 and GANC were selected, replicated, and expanded. The integrity of these clones was verified by Southern blot and DNA sequence analysis; chromosome counting

performed at the Animal Genomics Service of Yale University confirmed that all clones were euploid. Clones were then microinjected into C57BL/6 blastocysts to produce chimeric founders. Highly chimeric male founders were mated to C57BU6 females, and F1 pups were genotyped to identify those carrying the gene-targeted allele with the following primers: forward, 5′-ATAGCAGGCATGCTGGGGAT-3′ and reverse, 5′-GGCTCAGTAAAACACAGGCC-3′ (amplicon of 359 bp). After expansion of correctly identified F1 pups, the loxP-flanked Neo cassette was excised by breeding with mice expressing Cre recombinase (Jackson Laboratory). Mice were screened for Neo cassette removal with the following genotyping primers: forward, 5′-GGACAGACACTCTCCCAGTTAGTG-3′ and reverse, 5′-GGCTCAGTAAAACACAGGCC-3′ (amplicons of 1000 and 437 bp). A single founder was used to establish our Atase1⁻/⁻ colony with the aforementioned genotyping primers.

*Atase2⁻/⁻ mouse generation.* The mouse *Nat8* gene coding sequence was edited with CRISPR/Cas9 technology by the Genome Editing and Animal Models Core at the University of Wisconsin-Madison. One-cell fertilized, C57BL/6 embryos were injected with two gRNAs and Cas9 protein then transferred into pseudopregnant females. After birth and weaning, pups were genotyped and sequenced for the desired mutation. A single founder with the Nat8 protein sequence MAS-FRIRQNKX was used to establish our atase2⁻/⁻ colony with the genotyping primers as follows: forward 5′-AGAAGCTGGGTGGTGAGTAGAGGTTTAG-3′ and reverse 5′-GTCAGGAGTTGTGTGAACGGCATCAG-3′ (amplicons of 996 and 291 bp).

**Animals.** All animals used in this study are *Mus musculus* strain C57BL/6 J (MMRRC Stock No. 000664-JAX) with sex and age information listed in each figure legend. APP₆₉₅/swe/PS1-dE9 (APP/PS1) double transgenic mice were obtained from Jackson Laboratory (MMRRC Stock No. 34832-JAX) and crossed with our in-house Atase1⁻/⁻ and Atase2⁻/⁻ colonies. Mice were housed in standard cages provided by the University Laboratory Animal Resources and grouped with littermates, 1–5 mice per cage. Except for acetyl-CoA and CoA level measurements (described below), animals were supplied standard chow and water ad libitum. Animal experiments were performed in accordance with the National Institutes of Health Guide for the Care and Use of Laboratory Animals and were approved by the Institutional Animal Care and Use Committee of the University of Wisconsin-Madison (protocol #M005120).

**Behavior testing.** All behavioral testing was conducted within the Waisman Center Behavioral Testing Service (Madison, WI, USA). The experimenter was blind to the genotype of the mice during testing. All mice received a minimum of 30 min acclimation time to the testing room prior to each behavior assay. The following behavioral assays have been previously described: *novel object recognition*, *marble burying assay*, and *fear conditioning paradigm*[7].

*Open field exploration.* Open field exploration sessions lasted 30 min, and each mouse received 1 session. Each mouse was removed from its home cage and placed in the center of the arena. The Omnitech Fusion system used photobeams to continuously monitor and record the animal's placement during the session. Testing variables included total distance traveled and time spent in the center portion of the open field vs. the periphery (a measure of anxiety). Data were recorded using the Omnitech Fusion system with a center ratio zone map.

*Light/dark exploration.* Each mouse was placed into a split arena for a 10 min assay. Time spent in the dark portion of the arena and entries into the dark portion of the arena were recorded.

**Kidney function assessment.** Plasma urea nitrogen, plasma creatinine, urine albumin, and urine creatinine were measured by the UW Health Clinical

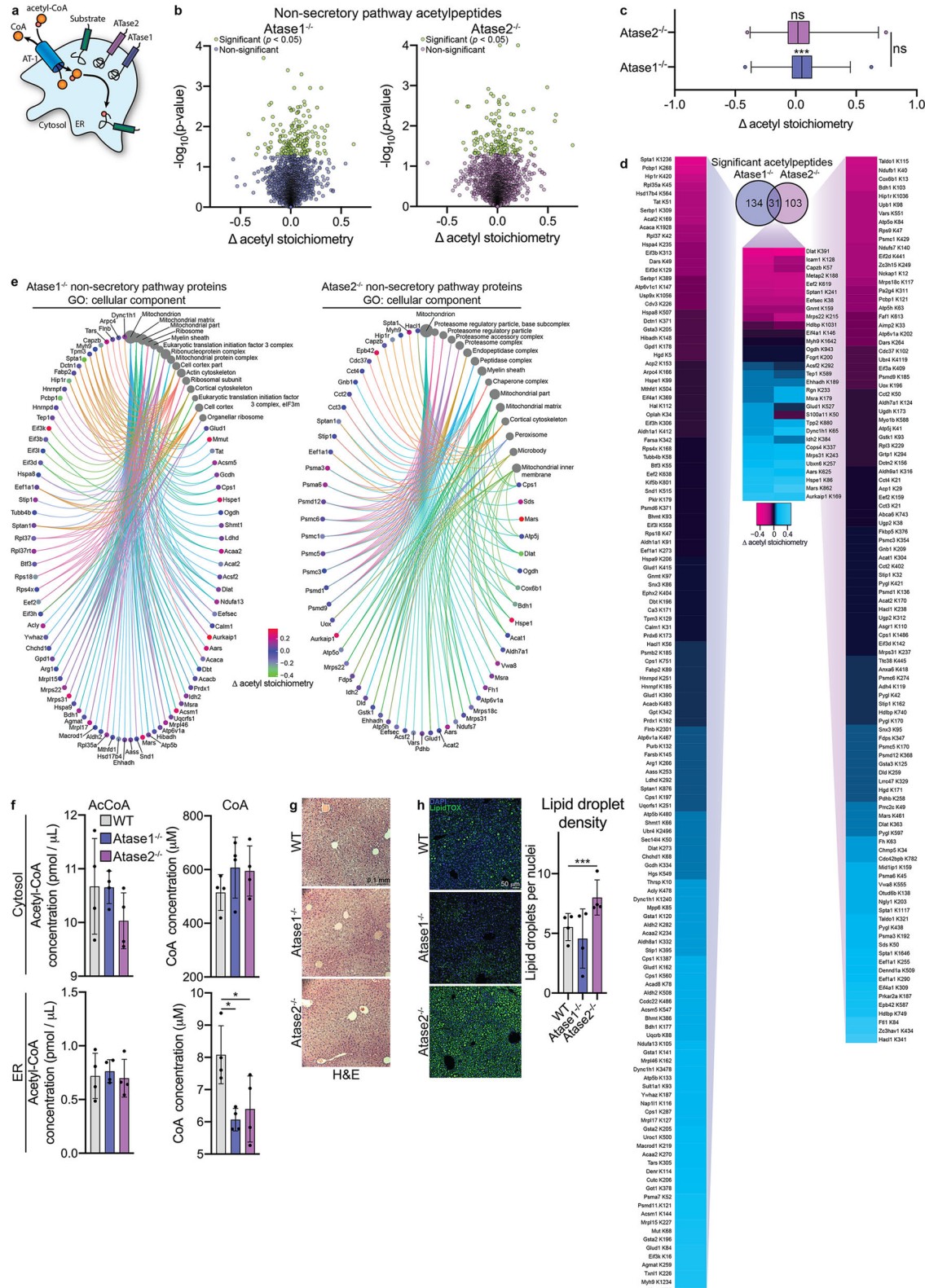

Laboratory (Madison, WI, USA). Blood was collected by transcardial puncture of $CO_2$-euthanized animals into a 0.5 M ethylenediaminetetraacetic acid (EDTA)-flushed needle and syringe. Samples were immediately centrifuged at $10,000 \times g$ for 5 min and the plasma supernatant was collected and flash frozen for storage at $-80\,^{\circ}C$ before testing. Spot urine was collected by scruffing animals on a hydrophobic surface and was sent immediately for laboratory testing. Kidney to body mass ratios were calculated by weighting the total carcass mass and wet mass of left and right kidneys separately after $CO_2$ euthanasia.

### Cell culture

*Mouse embryonic fibroblasts (MEFs) preparation and culturing.* MEFs were prepared from timed pregnant females on embryonic days E12.5–E14.5. After $CO_2$ euthanasia of the pregnant female, the embryo head and visceral organs were removed, body minced in cell culture grade trypsin-EDTA (0.25%; Gibco™; #25200056), and incubated at $37\,^{\circ}C$ for 15 min. Trypsin was quenched by addition of complete media containing Dulbecco's modified Eagle's medium (DMEM; Corning, #10–017-CV) supplemented with 10% Fetal Bovine Serum (FBS; Corning,

**Fig. 6 Atase1$^{-/-}$ and Atase2$^{-/-}$ exhibit differential adaptations from impaired At-1 acetyl-CoA/CoA antiporter activity. a** Schematic diagram of the ER-based acetylation machinery. The AT-1 antiporter brings in acetyl-CoA from the cytosol in exchange for CoA. Newly folded proteins within the ER lumen are acetylated by ATase1 and ATase2, creating free CoA as a byproduct. **b** Volcano plots of lysine acetylation stoichiometry from enriched liver ER from WT, Atase1$^{-/-}$, and Atase2$^{-/-}$ mice. Only acetylpeptides that are not identified with a subcellular localization of "ER," Golgi," or "secreted" are included. **c** Comparison between significantly changed acetylpeptides shown in panel a. Box plots display 25–75th percentile with a line at the mean value, whiskers at the 1st and 99th percentiles, and dots representing data points outside the 99th percentile. Statistical testing between the Atase1$^{-/-}$ and Atase2$^{-/-}$ distributions was conducted via the Kolmogorov–Smirnov test; ns not significant. Labels over the mean value designate results of a one-sample *t*-test comparing to a mean value of 0, ***$p < 0.0005$; ns not significant. **d** Overlap between significantly changed acetylpeptides shown in (**a**) with a heatmap showing the stoichiometry of lysine acetylation changes from WT. **e** Gene ontology (GO) overrepresentation analysis of the non-secretory pathway proteins harboring the acetylation sites that were significantly changed from WT as shown in (**b**). The dot size of each network category is scaled by number of proteins included in that category. The top 15 enriched categories are shown that have been filtered by a false discovery rate threshold of $q = 0.05$. **f** Liver ER and cytosolic fractions were assayed for acetyl-CoA and coenzyme A from mice that underwent a 6 h fast. Data are mean ± SD, $n = 4$ male mice per genotype. Mice were 3 months of age at time of study. *$p < 0.05$ via one-way ANOVA with Dunnett's multiple comparison test. **g** Liver H&E staining on 5 µm paraffin-embedded sections. **h** Liver frozen sections (10 µm) stained for nuclei (DAPI) and lipid droplets (LipidTOX). Data are mean ± SD, $n = 4$ male mice per genotype. ***$p < 0.0005$ via two-way ANOVA (animal vs. genotype) with Dunnett's multiple comparison test. For (**g**, **h**), mice are 6–9 months of age at time of study.

#35-010-CV) and 1% penicillin/streptomycin (Gibco$^{TM}$, #10378016). After tissue dissociation via pipetting, debris was allowed to settle, and the remaining cell suspension was plated in complete media and cultured in a 37 °C humidified incubator with 5% $CO_2$. Confluent cells were dissociated with trypsin-EDTA and frozen in DMEM supplemented with 20% FBS and 10% cell culture grade DMSO. For experiments, MEFs were cultured in complete media in a 37 °C humidified incubator with 5% $CO_2$ and used up to three passages.

*Nucleofection.* Confluent MEFs were dissociated with trypsin-EDTA, diluted in complete media, and counted by hemocytometer. Between 1 and $8 \times 10^6$ cells were centrifuged $100 \times g$ for 10 min and resuspended in 800 µL filter-sterilized nucleofection solution (18 parts 120 mM sodium phosphate buffer pH 7.2, 1 part 100 mM KCl, 1 part 300 mM $MgCl_2$, and 4.5 parts 50 mM mannitol) supplemented with vector DNA (5 µg per $1 \times 106$ cells). The cell suspension was transferred to an electroporation cuvette (Bulldog Bio; # 12358-347) and nucleofected in the Amaxa Nucleofector II Device (Lonza) using the U-023 program. Immediately after, the cell suspension was diluted with RPMI-1640 media (Gibco$^{TM}$; # 11875101) and incubated for ≥5 min in a 37 °C humidified incubator with 5% $CO_2$ before plating. Cells were incubated 24–48 h before analysis.

*Lipofectamine$^{TM}$ transfection.* MEFs were grown to 60–80% confluency and transfected with Lipofectamine$^{TM}$ Stem Transfection Reagent (Invitrogen; #STEM00015) according to kit instructions. Cells were incubated 24–48 h before analysis.

*Transduction.* MEFs were grown to 60–80% confluency and transduced with BacMam 2.0 Premo$^{TM}$ Autophagy Sensor LC3β-GFP (Invitrogen; #P36235) according to kit instructions. Cells were cultured for 20–24 h before imaging.

**Liver ER enrichment and immunoprecipitation.** Enriched liver ER was prepared as previously described[17] using the Endoplasmic Reticulum Enrichment Extraction Kit (Novus; # NBP2-29482). For immunoprecipitation, the total ER pellet was resuspended in kit-provided 1X suspension buffer supplemented with protease inhibitor cocktail and 1% Triton X-100. For co-immunoprecipitation, no detergent was added to the final suspension buffer. Immunoprecipitation and co-immunoprecipitation was performed with 500–1000 µg enriched liver ER using anti-acetylated lysine (Cell Signal Technologies; #9441; 1:100) or anti-ATG9A (abcam; #ab108338; 1:100) primary antibodies as previously described[12].

**Subcellular fractionation.** Nuclear protein was extracted from liver using the Nuclear Extraction Kit (abcam; ab113474) according to kit instructions.

**Western blotting.** Western blotting was conducted as previously described[1,8,15]. The following primary antibodies were used in this study: anti-ATG9A (abcam; #ab108338; 1:1,000), anti-FAM134B (abcam; ab151755; 1:1,000), anti-SEC62 (abcam; ab140644; 1:1,000), anti-LC3β (Cell Signal Technologies; #2775; 1:500), anti-Beclin (Cell Signal Technologies; #3738S; 1:1,000), anti-p62/SQSTM1 (Sigma-Aldrich; P0067-200UL; 1:1,000), anti-β-actin (Cell Signal Technologies; 3700 or 4967; 1:1,000 to 1:5,000), anti-p-PERK (Santa Cruz; sc-32577; 1:200), anti-PERK (Cell Signal Technologies; #3192; 1:1,000), anti-eIF2α (Cell Signal Technologies; 9722; 1:1,000), anti-p-eIF2α (Cell Signal Technologies; 9721; 1:1,000), anti-p-IRE1 (Novus; #NB100-2323; 1:500), anti-IRE1 (Cell Signal Technologies; #3294; 1:1,000), anti-ATF6 (Millipore; #09-069; 1:250), anti-BiP/GRP78 (Cell Signal Technologies; #3177; 1:1,000), anti-ATF4 (Cell Signal Technologies; #11815; 1:1,000), anti-H3 (Active Motif; #39763; 1:10,000), anti-GFAP (Agilent; #Z0334; 1:1,000), anti-IBA1 (abcam; ab178847; 1:1,000), and anti-α-SMA (Sigma-Aldrich; #A7607; 1:100).

Donkey anti-rabbit or goat anti-mouse IRDye 800CW and 680RD-conjugated secondary antibodies (LI-COR Biosciences, #'s 925-32213, 925-32210, 926-68073, 926-68070) were used for infrared imaging (LICOR Odyssey Infrared Imaging System; LI-COR Biosciences). For co-immunoprecipitation experiments, mouse anti-rabbit TrueBlot HRP-conjugated secondary antibody (Rockland; #18-8816-31) was used followed by chemiluminescent detection with Amersham ECL Western Blotting Detection Kit (GE Healthcare; #GERPN2209) on the Azure c600 imager (Azure Biosystems). For enriched liver ER Western blotting, target proteins were normalized to total protein staining performed before immunodetection (LiCor; #926-11010). The original uncropped Western blot images included in the manuscript can be found in Supplementary Figs. 7–12.

**Immunocytochemistry.** For structured illumination microscopy (SIM), MEFs were plated on #1.5 glass coverslips (Harvard Bioscience, Inc.; #64-0712) and processed as previously described using Lipofectamine$^{TM}$ Stem Transfection Reagent[17]. For ER/Lc3β-labeled confocal imaging, MEFs were nucleofected with pLenti-X1-hygro-mCherry-RAMP4 plasmid, a gift from Jacob Corn (Addgene plasmid #118391)[44], and plated on #1.5 glass coverslips. After 24–48 h, cells were processed for endogenous Lc3β staining as previously described using anti-LC3β (Cell Signal Technologies; #2775; 1:200)[44]. Cells were imaged using a Nikon A1 confocal microscope with a 100× oil immersion lens (NA = 1.4). Z-stack images (1024 × 1024 pixels at 0.12 µm/pixel with 10 z-stacks at 0.3 µm step size) were acquired using NIS-Elements AR version 5.11.01 software with 405 nm (blue channel), 488 nm (green channel), and 561 nm (red channel) laser wavelengths at a pinhole size of 67.7 µm. Images were analyzed on Imaris version 9.5 (Bitplane) using voxel-by-voxel co-localization of the green and red channels, and the Pearson's correlation coefficient in the dataset volume was reported.

**Live cell imaging**

*ER tandem reporter (EATR).* MEFs were plated on black 96-well plates with #1.5 glass bottom (Cellvis; # P96-1.5H-N) at a density of 40,000 cells/well. Cells were transfected with TetOn-mCherry-eGFP-RAMP4 plasmid, a gift from Jacob Corn (Addgene #109014)[44], using Lipofectamine$^{TM}$ Stem Transfection Reagent and maintained in 4 µg/mL doxycycline (Sigma-Aldrich; #D9891). After 24 h, media was changed to the following conditions: fed, DMEM without phenol red (Gibco$^{TM}$; #21063-029) + 10% FBS + 1% penicillin/streptomycin; starved, Earl's buffered saline solution (EBSS; Gibco$^{TM}$; #14155-063); starved with folimycin, EBSS + 100 nM folimycin (Millipore; #344085). After 4 h, cells were imaged in a live cell imaging chamber (37 °C with 5% $CO_2$) on a Nikon A1 confocal microscope using a 100× oil immersion lens (NA = 1.4). Single z-slice images (1024 × 1024 pixels; 0.12 µm/pixel) were acquired using NIS-Elements AR version 5.11.01 software with 488 nm (green channel) and 561 nm (red channel) laser wavelengths at a pinhole size of 67.7 µm. Images were analyzed on Imaris version 9.5 (Bitplane) by performing a red – green channel subtraction and quantifying the number of puncta using a surface reconstruction.

*Autophagosome trafficking.* MEFs were nucleofected with pGFP-LC3, a gift from Tamotsu Yoshimori (Addgene #21073)[53], and plated on a glass bottom 35 mm dishes (MatTek; #P35G-1.5-14-C) or plated and transduced with BacMam LC3β-GFP as described above. After 20–24 h, cells were imaged live using the Andor Revolution XD spinning disc microscopy system as previously described[54].

**Histology and immunostaining.** Histology and immunostaining techniques were performed as described previously[8,14,15]. For thioflavin-S staining, deparaffinized and rehydrated slides were incubated for 10 min in 1% thioflavin-S (Sigma-Aldrich; #T1892-25G) dissolved in 50% ethanol. Slides were rinsed in 80% ethanol and 50%

ethanol for 1 min each, briefly rinsed in distilled water, and mounted with aqueous mounting media with DAPI (Electron Microscopy Sciences; #17985-50). Picrosirius red staining was performed according to kit instructions (abcam; #ab150681). Liver LipidTOX staining was performed as previously described[2]. The following primary antibodies were used: anti-GFAP (Thermo Fisher; # PIMA512023; 1:1,000), anti-IBA1 (Abcam; #ab178847; 1:1,000), anti-NeuN (EMD Millipore; #ABN91MI; 1:1,000), anti-synaptophysin (abcam; #ab32127; 1:200), anti-PSD95 (Thermo Fisher; #MA1-045; 1:200), and anti-collagen I (abcam; #ab34710; 1:50). For actin immunostaining, phalloidin-TRITC (Sigma-Aldrich; # P1951-.1MG; 1:1,000) was used during the secondary antibody incubation step.

Bright-field images were acquired using a Leica DM4000 B microscope with a 10× or 20× air objective using Image-Pro version 6.3. All fluorescently-labeled slides were imaged on a Nikon A1 confocal microscope using NIS-Elements AR version 5.11.01 software with 405 nm (blue channel), 488 nm (green channel), 561 nm (red channel), and 640 nm (far red) laser wavelengths. For collagen/actin-stained pancreas slides, single z-slice images (1024 × 1024 pixels; 0.63 μm/pixel) were acquired using a 20× air objective (NA = 0.75) at a pinhole size of 255.4 μm. For thioflavin-S and Gfap/Iba1/NeuN-stained slides, single z-slice images (1024 × 1024 pixels; 1.24 μm/pixel) were acquired using a 10× air objective (NA = 0.3) at a pinhole size of 129.0 μm and 255.4 μm, respectively. For synaptophysin/Psd-95/NeuN-stained slides, z-stack images (1024 × 1024 pixels at 0.21 μm/pixel with 15 z-stacks at 0.2 μm step size) were acquired on a 60× oil immersion objective (NA = 1.4) at a pinhole size of 39.6 μm. For liver LipidTOX slides, single z-slice images (1024 × 1024 pixels; 0.62 μm/pixel) were acquired using a 20× air objective (NA = 0.75) at a pinhole size of 20.4 μm.

Plaque analysis on thioflavin-S images was performed in ImageJ version 2.0 by making binary images via an intensity threshold and counting objects using the particle analyzer. Synapse loss was quantified on synaptophysin/Psd-95 fluorescence images using Imaris version 9.5 (Bitplane) by creating 1 μm-diameter red and green spots and counting the number of co-localized spots within 1 μm of each other.

**Luciferase reporter assay**. MEFs were co-nucleofected with p5xATF6-GL3 plasmid, a gift from Ron Prywes (Addgene #11976)[55], and pNL1.1 PGK [Nluc/PGK] internal control plasmid (Promega; # N1441) in a 50:1 mass ratio and plated on white 96-well plates with clear bottoms (Corning; #3610). After 24 h, nanoluciferase and luciferase activities were measured using a dual luciferase kit (Promega, #N1610) on a GloMax plate reader (Promega). Normalized activity was calculated as the luciferase:nanoluciferase ratio and represented as the fold change over WT.

**Mercapturic acid assay**. Spot urine was collected by scruffing animals on a hydrophobic surface and flash frozen for storage at −80 °C before testing. Samples were assayed for 1,4-dihydroxynonane mercapturic acid (DHN-MA) by ELISA (Bertin Pharma; #A05033) following kit instructions using a sample dilution of 1:100. The absorbance at 410 nm was measured 120 min after the addition of Ellman's reagent on a VersaMax microplate reader (Molecular Devices).

**Acetyl-CoA and Coenzyme A assays**. Mice were fasted by removing chow at the beginning of the light cycle (06:00) and $CO_2$-euthanized after 6 h (12:00). Liver was rapidly removed and diced into small pieces, rinsed in PBS, and flash frozen. Enriched liver ER was prepared in the same manner as above with the final centrifugation supernatant being the cytosolic fraction. The ER pellet was resuspended in 500 μL acetyl-CoA assay buffer and cytosolic fraction mixed 1:1 with acetyl-CoA assay buffer (Abcam; #ab87546). Both fractions were deproteinized (BioVision; #K808) according to kit instructions. Acetyl-CoA concentrations were determined with an acetyl-CoA assay kit (Abcam; #ab87546) according to kit instructions using the standard curve. Fluorescence was measured on a GloMax plate reader (Promega) using a green fluorescence filter (Ex = 525 nm; Em = 580−640 nm). Interpolated concentrations (in pmol/well) were converted to pmol/μL (based on adding 50 μL of sample per well) then scaled by a dilution factor calculated from the deproteinization protocol (2.5 for cytosolic fraction and 1.25 for ER fraction). Coenzyme A (CoA) concentrations were determined with a CoA assay kit (Abcam; #ab138889) according to kit instructions using the standard curve. Fluorescence was measured on a GloMax plate reader (Promega) using a blue fluorescence filter (Ex = 490 nm; Em = 510−570 nm). Interpolated concentrations (in μM) were scaled by a dilution factor calculated from the deproteinization protocol (125 for cytosolic fraction and 1.25 for ER fraction).

**Reverse transcription quantitative PCR (RT-qPCR)**. RNA extraction, cDNA synthesis, and RT-qPCR was performed as previously described[14,17,18]. For absolute quantification, $log_{10}$(number of gene copies) vs. PCR cycle standard curves were generated using plasmid DNA. Nat8-pCMV6 (Origene; #MR202704) was used for *Atase2* mRNA quantification. For *Atase1* mRNA quantification, mouse *Atase1* was subcloned into the Nat8-pCMV6 plasmid by removing the *Nat8* ORF via restriction enzyme digestion (SgfI/MluI) and ligating in the *Nat8b* ORF amplicon generated from the following primers via traditional PCR with Phusion DNA polymerase (Thermo Fisher; #F531S) on a WT mouse tail lysate: forward 5′-GCCGCCGCGATCGCCATGCCTAGATTTGAG-3′ and reverse 5′-TACTC

CTTCCCTTCTGCCACGCGTACGCGG-3′ (66 °C annealing temperature). The Nat8b-pCMV6 construct was verified by DNA sequencing. The number of gene copies for unknown cDNA samples is reported as number of copies per ng RNA added to cDNA reaction. Primer sequences not previously reported for RT-qPCR are as follows: mouse *Aft4* forward 5′-ATGGCCGGCTATGGATGAAT-3′ and reverse 5′-CGAAGTCAAACTCTTTCAGATCCATT-3′; mouse *BiP/Grp78* forward 5′-CTGAGGCGTATTTGGGAAAGAA-3′ and reverse 5′-TGACATT CAGTCCAGCAATAGTG-3′.

**Stoichiometry of acetylation**. Enriched liver ER was prepared in the same manner as above with *n* = 4 animals per genotype and was analyzed as previously described[2,45]. Acetyl stoichiometry values are reported as a range from 0 to 1 representing 0–100% of a detected lysine site being endogenously acetylated, respectively. In this study, changes in acetyl stoichiometry values (Δ acetyl stoichiometry) are always reported as the Atase knockout value minus the WT value. The stoichiometry of an acetylation site was determined to be 0 if these conditions were met: (1) it had measurable stoichiometry in at least 3 of 4 control samples, but not in the experimental samples, and (2) we could detect only the heavy-labeled peptide (in vitro acetylation) but no light-labeled peptide (endogenous acetylation) in at least 3 of 4 experimental samples. Subcellular localization was annotated by Uniprot, Human Protein Atlas, and GO Terms; if there were discrepancies in localization between databases, secretory pathway-associated localizations ("ER", "Golgi", or "secreted") were used. Pathway analysis and network plot construction was conducted using the R package enrichplot implementing an overrepresentation analysis (ORA) with the *Mus musculus* organism database. Relevant parameters included a minimum and maximum number of genes for a category of 5 and 2000, respectively, with the Benjamini–Hochberg method for multiple test adjustment to a false discovery rate of 0.05.

**Statistics and reproducibility**. No statistical method was used to determine necessary sample size for each experiment. The minimum number of replicates for all experiments were *n* = 3, which represent either the number of mice per genotype or mouse embryonic fibroblast lines per genotype (derived from independent embryos). The only exception is the survival analysis for which we have determined requires at least 40–50 animals per group[14,15,17]. Data analysis was performed using GraphPad Prism version 8.4.3. Data are expressed as mean ± SD unless otherwise specified. Comparison of the means was performed using two-tailed Student's *t* test for two groups and ordinary one-way or two-way ANOVA for ≥3 groups followed by either Tukey–Kramer (comparison between all groups) or Dunnett's (comparison to one control group) multiple comparisons test. Statistical test details are described in the figure legends. Grubb's test was used to remove outliers. Differences were declared statistically significant if *p* < 0.05, and the following statistical significance indicators are used: *$p$ < 0.05; **$p$ < 0.005; ***$p$ < 0.0005.

**Reporting summary**. Further information on research design is available in the Nature Research Reporting Summary linked to this article.

## Data availability
All data included in this study are stored and maintained by the corresponding author and are available from the corresponding author upon reasonable request. Source data underlying graphs shown in figures and supplementary figures are provided in Supplementary Data 1. The acetyl-proteomics data that support the findings of this study have been deposited to the ProteomeXchange Consortium (ID number PXD023641) and the MassIVE partner repository (ID number MSV000086712).

## Code availability
The R script that was used to process the acetyl-proteomics data have been deposited on Github with the identifier (https://doi.org/10.5281/zenodo.4447491). The README file found on Github describes how the input data for the scripts can be accessed through ProteomeXchange accession code PXD023641.

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

## Acknowledgements

This research was supported by NIH (NS094154, AG053937, AG057408, EY023299 and GM065386), the Department of Veterans Affairs (I01 BX004202), a core grant to the Waisman Center from NIH/NICHD-U54 HD090256, and a core grant for vision research to the University of Wisconsin-Madison from NIH/NEI-P30 EY016665. M.J.R. was a recipient of the Wisconsin Distinguished Graduate Fellowship. We would like to thank Dustin Rubinstein and Kathy Krentz at the Genome Editing & Animal Models Core of UW-Madison for the generation of Atase1$^{-/-}$ and Atase2$^{-/-}$ mice; Ruth Sullivan for pathology analysis; Heather Mitchell at the UW-Madison Waisman Behavior Testing Service for conducting mouse behavior testing; and S. Shapiro, B. Sheehan, C. Galvan, and Y. Peng for technical assistance.

## Author contributions

M.J.R., A.J.L., G.K., V.C.B., and W.E.K. performed the experiments and analyzed the data. A.L., J.M.D., and L.P. provided critical advice for the experiments. L.P. designed the overall study. M.J.R. and L.P. wrote the paper with input from all authors.

## Competing interests

The authors declare the following competing interests: J.M.D. is a co-founder of Galilei BioScience Inc and a consultant for Evrys Bio. Remaining authors have no competing interests to disclose.
