## [Peer Review File · Communications Biology]

Reviewers' comments:

Reviewer #1 (Remarks to the Author):

The manuscript by Rigby MJ et al., describes the characterization of mice with deletion in one of two ER acetyltransferases and the adaptive changes in reticulophagy, macroautophagy, protein acetylation and acetyl-CoA metabolism that follow these genetic manipulations. The authors use a multitude of in vivo and in vitro approaches to evaluate organ function and the above processes. Furthermore, they evaluate the ability of Atase1 or Atase2 deletion to ameliorate the proteotoxicity phenotype of a mouse model of Alzheimer's disease, which could have potential translational implications for the treatment of this disease.

The manuscript is overall well written and the conclusions are, for the most part, supported by robust, convincing data. The experimental methods are described in sufficient detail to enable reproduction of the work and the statistical analysis of the data is appropriate.

Main comments:

- 1) The authors use liver lysates and MEFs derived from their Atase1 and Atase2 KO mice to investigate the potential activation of reticulophagy. They found evidence of modestly elevated reticulophagy in the Atase1 KO lysates/MEFs but not in the Atase2 KO lysates/MEFs, and they suggest that this could be due to a compensatory increase in Atase1 expression in the Atase2 KO liver/MEFs. It is unclear why the authors do not use western blotting to quantify expression of Atase1 protein in the Atase2 KO systems that they use. Furthermore, while the mRNA levels of Atase1 tended to be higher in the Atase2 KO livers, the difference was not statistically significant (Fig 1g) and no Atase1 mRNA levels data were provided for the Atase2 KO MEFs. Therefore, the conclusion that elevated Atase1 levels may be compensating for the deletion of Atase2 is currently not supported by the data provided, but it is an important point to address.
- 2) It is understood that transfections and cell imaging are more easily done in MEFs, but the authors keep mixing results from MEFs and liver lysates without the appropriate controls (see also above). For example, most of the markers of the autophagic flux are quantified in KO MEFs (Fig. 3a), but the authors selectively use liver lysates to show a decrease in p62 levels that was not observed in the MEFs (Fig. 3b). What happens to the levels of Beclin and LC3b-I and II in the livers of the KO mice?
- 3) The authors should comment on potential mechanism through which BiP, but not processed ATF6, is increased in Atase2 KO mice. Furthermore, the luciferase data in Fig 3g are highly variable and would benefit from the addition of 1-2 samples/group.
- 4) While deletion of Atase2 in APP/PS1 female mice fails to increase their life span, the percentage of male mice that survive at 10 months of age and the number of synaptophysin and Psd-95 co-localized puncta is similar between APP/PS1;Atase1^{-/-} and APP/PS1;Atase2^{-/-} mice. Therefore the conclusion that Atase1 deletion confers a more robust rescue (page 24, line 541) is only true for females and needs to be more specific in the text. Furthermore, the authors could elaborate/speculate a little more on potential mechanisms leading to male/female differences in APP/PS1;Atase2^{-/-} mice, but not APP/PS1;Atase1^{-/-} mice.
- 5) In discussing the data in Figs 5d and e, the authors should indicate whether the localization of the proteins and/or acetylation sites they identified are consistent with the orientation (ER lumen) of the active sites of Atase1 and 2.
- 6) The acetyl-CoA and CoA data reported in Fig 6f are expressed as μM or $\text{pmoles}/\mu\text{l}$. If these values refer to actual concentrations in the cytosol and ER lumen, how were the volumes of these subcellular compartments calculated?
- 7) If the deletion of Atase1 or 2 blunts the AT1-mediated import of acetyl-CoA into the ER lumen due to the decreased production of free CoA, then accumulation of acetyl-CoA in the cytosol (even if transient, as it was not detected) should lead to similar changes in the acetylome of both KO mice. Instead, only 31 acetylpeptides were common to the two mouse strains and a large number of categories in Fig 6e were different between KO mice. The authors should expand the discussion of these results with the goal of providing an explanation. Specifically, do they expect the amount of

acetyl-CoA 'accumulating' in the cytosol of the Atase1 KO and Atase2 KO mice to be different? And if this is the case, why?

8) In fasted mice, cytosolic acetyl-CoA is unlikely to be diverted to fatty acid synthesis, as the authors seem to suggest at page 28, lines 631-632, or in the discussion, and the accumulation of lipid droplets in the Atase2-/- mice could be due to a decrease in fatty acid oxidation or impaired VLDL secretion. The authors should evaluate, or at least discuss, how the proteins identified in Fig 6e may be connected to lipid droplet accumulation in the liver in the fasted state (the use of fasted mice is indicated at page 28, line 630).

Minor

Mouse gene names should be italicized in the manuscript and figures.

Reviewer #2 (Remarks to the Author):

The manuscript is implicating important aspects of Atase1 and Atase2 metabolism. The work done on knockout mice is well designed. The results are going to augment the knowledge regarding AcetylCoA regulation in the ER. The manuscript also presents results that are different from some previously reported findings. Although, the manuscript raises issues in APP/PS1 AD system, the abstract and conclusion need to be more elaborate on this. The involvement of Atase2 in protein quality control through the activation of ER stress via ATF6 up-regulation is a significant finding. The results of the research would be valuable for further investigation.

Specific comments:

1. Correction in Line 92: 'cell degradation system' in place of cell degrading system
2. Correction in Line 93: replace disposing with 'disposal'.
3. Correction in Line 94: omit 'across lifespan'.
4. Correction in Line 94: replace and with 'whereas'.
5. Correction in Line 102: replace but differential changes with 'while differential changes'.
6. In figure 1b: the gel for heterozygous Atase2 is showing two equal intensity bands around 1000bp, please clarify.
7. In figure 1c: check the labeling of marker sizes
8. Line405 should be: kidney weight, plasma creatinine and urea nitrogen,
9. Figure 1 (Line 755): All mice under the study are male, it would be significant to observe the same events in equal number of female mice.
Figure 1c doesn't have caption.
Line 397: data should be shown for the necropsy and histologic assessment of organs.
10. Line 408: data should be included
11. Line 440: the immunocytochemistry on Atase1-/- knockout mice MEFs show decrease in Fam134b puncta (figure2b).
But the literature says "downregulation of the ER-resident protein FAM134B prevents autophagic clearance of the ER and results in ER expansion" Abhilash I. Chiramel, Jonathan D. Dougherty, Vinod Nair, Shelly J. Robertson, Sonja M. Best, FAM134B, the Selective Autophagy Receptor for Endoplasmic Reticulum Turnover, Inhibits Replication of Ebola Virus Strains Makona and Mayinga, The Journal of Infectious Diseases, Volume 214, Issue suppl_3, October 2016, Pages S319-S325, <https://doi.org/10.1093/infdis/jiw270>
- Line 441: Also, the manuscript shows an increase in Lc3beta, which activates reticulophagy.
Please explain this differential observation.
12. Figure 2 (Line757) : 'Knockout of Atase1 display activated reticulophagy', the results shown here are with respect to enriched liver ER. Data should be included for brain

samples as well.

13. Line 522: 'most notable in the hippocampus; we only...

This should be written as 'most notable in the hippocampus; whereas we only observed...

14. Line 822: All the results are shown only in male mice, data from female subjects should be compared.

15. Line 541: 'with knockout of Atase1 providing a more robust rescue'. But from the data represented in Figure 4, it's not really significant to assert here that Atase1-/- phenotype is a robust rescuer.

16. Figure 5 c should include proper labelling, the caption 'Genes' should be added on the heatmap.

17. Line 679: replace 'of the differing' with 'about the different'.

18. Line 682: Omit the line 'Much of the consequence.....to consider'.

19. Line 694: Replace 'while' with 'whereas'

Add 'full stop' after 'did not'. Rewrite as 'Additionally, Atase1-/- mice(omit 'see Atase1-/- volcano plot in').

20. Line 696: replace 'assume' with 'presume'

21. Line 697: add 'to' build..

22. Line 707: add 'only' before 'one Atase..

Reviewer #3 (Remarks to the Author):

The work entitled "Endoplasmic reticulum acetyltransferases Atase1/Nat8b and Atase2/Nat8 differentially 1 regulate reticulophagy, macroautophagy, and cellular acetyl-CoA metabolism" by Rigby et al, describes the characterization of mice lacking ATase1 and ATase2, the only known acetyltransferases devoted to lysine acetylation in the ER lumen. The mice have apparently no major phenotype. However, at cellular levels the lack of ATase proteins alter global protein acetylation, ER homeostasis, promote reticulophagy and macroautophagy.

The authors nicely show that the deletion of Atase1 in mouse model of Alzheimer disease reduced plaque density and neuroinflammatory signs and as consequence improve survival. There is no explanation for these results.

Lastly the authors used mass spectrometry to characterize substrate acetylation in vivo using ER fractions isolated from livers of WT, Atase 1 KO, and Atase 2 KO mice. They found about 80 proteins with altered acetylation in KOs compared to wt.

In addition they demonstrated that deletion of Atase proteins affect general acetylation and liver lipid metabolism.

In summary, this work contains several data, both in vitro and in vivo, obtained by using multiple approaches. It is clearly written, and easy to follow.

The observation that there is an activation of reticulophagy, as well as of macroautophagy in Atases KO cells is consistent with an increased ER stress due to alteration of protein acetylation.

I am not convinced that this is due to a specific regulation of ATG9 by ATASE. The authors did not provided any evidence showing that ATG9a hypoacetylation is triggering activation of reticulophagy via direct interaction with FAM134B. To convincingly support this hypothesis much more data will be needed and this will be far beyond the scope of this paper. In addition, immunoprecipitations experiments did not show negative controls and inputs are not always showing same amount of proteins. I suggest the authors to remove data 2d-f as well as the related text from the manuscript.

Western blotting analysis in fig 3 show that samples were not loaded on the same gel. If this is the case the authors should repeat the experiments loading samples on the very same gel, otherwise the quantification is useless since different exposures will affect the results.

Figure 5: Fig c, d, e, and 6d,e: I think these two figures need to be reorganized, the text is too small. I also suggest the authors to move these figures and related text in the first part of the manuscript, after the description of the atase mouse phenotype. Indeed these proteomic analysis represent the vivo investigation of Atase protein functions.

POINT-BY-POINT RESPONSE

We wish to thank the Editor and the Reviewers for their positive comments and suggestions. A comprehensive point-by-point response can be found below.

Reviewer #1

1) *It is unclear why the authors do not use western blotting to quantify expression of Atase1 protein in the Atase2 KO systems that they use. Furthermore, while the mRNA levels of Atase1 tended to be higher in the Atase2 KO livers, the difference was not statistically significant (Fig 1g) and no Atase1 mRNA levels data were provided for the Atase2 KO MEFs. Therefore, the conclusion that elevated Atase1 levels may be compensating for the deletion of Atase2 is currently not supported by the data provided, but it is an important point to address.*

Response:

Western blotting. Unfortunately, the currently available anti-NAT8 antibodies do not recognize mouse Atase1/Atase2. This was verified by transfecting HEK293 (human embryonic kidney) cells with mouse Atase1-myc-DDK or mouse Atase2-myc-DDK (see **Figure A** below; NOTE: if detected by both antibodies, mouse Atase would be yellow in the far right image).

FIGURE A

Liver variability of RT-PCR data. The variability in the RT-PCR data is quite common for expression studies in mouse tissue. The mean fold change in liver *Atase1* levels observed in *Atase2*^{-/-} mice vs. WT is 1.89 with a standard deviation of 1.40 ($p = 0.13$ via Student's *t*-test). By themselves (out of context), these numbers would suggest no statistical difference. However, when taken together, the data across the 7 tissue types examined, clearly suggest a broad attempt of the organism to compensate for the absence of *Atase2*.

Although we currently believe that the compensatory increase in *Atase1* expression observed in *Atase2*^{-/-} mice is the most likely explanation for the difference in phenotype between *Atase1* and *Atase2* KO mice, we do agree that it might not be the only one. We feel that the language used in the manuscript (...*The compensatory upregulation of Atase1 in the Atase2*^{-/-} mouse may play

a role in the phenotype observed below....) is already quite balanced and we would rather not engage in possible speculations.

2) It is understood that transfections and cell imaging are more easily done in MEFs, but the authors keep mixing results from MEFs and liver lysates without the appropriate controls (see also above). For example, most of the markers of the autophagic flux are quantified in KO MEFs (Fig. 3a), but the authors selectively use liver lysates to show a decrease in p62 levels that was not observed in the MEFs (Fig. 3b). What happens to the levels of Beclin and LC3b-I and II in the livers of the KO mice?

Response:

We have so far generated 5 mouse models of increased and decreased ER acetylation (published: AT-1^{S113R/+}, AT-1 nTg, AT-1 sTg; current paper: Atase1^{-/-} and Atase2^{-/-}). We have never observed striking tissue- or organ-differences. This is not surprising since every cell requires a fully functioning secretory pathway and fully functioning proteostatic mechanisms. Obviously, as we reported multiple times, cells and tissues that rely more heavily on the efficiency of the secretory pathway (such as neurons and hepatocytes) are more affected by our genetic manipulations. This is clearly reflected by the phenotype of the animals.

The fact that we are studying events that are fundamentally important for every cell of the organism facilitates our work because it allows us to use multiple/different tissues and primary cells according to the scientific question and experimental needs. For example, as the Reviewer points out, transfection and imaging are more easily performed with MEFs in culture. Similarly, the liver is congenial when isolation or enrichment of the rough ER is necessary. Furthermore, cells can be more easily controlled (i.e.; plated at the same time; same passage; same media; etc.) than mice.

However, to comply with the Reviewer’s request, we have looked at levels of Beclin, LC3b-I, and LC3b-II in the liver (see **Figure B** below). The results are in line with the MEF data. Specifically, we observed a significant increase in both LC3B-I and Beclin. The apparent different behavior of LC3B-I (increased) and LC3B-II (no change) is well documented in tissues of animals with chronic activation of autophagy and, therefore, not surprising (this complex behavior of LC3B-I/II is well discussed in the 2016 Autophagy guidelines, which we helped framing – please, see Autophagy 2016; 12: 1. The new guidelines are currently In Press and should be available soon).

FIGURE B

3) *The authors should comment on potential mechanism through which BiP, but not processed ATf6, is increased in Atase2 KO mice. Furthermore, the luciferase data in Fig 3g are highly variable and would benefit from the addition of 1-2 samples/group.*

Response:

As requested, we added more samples and updated the results (see new Figure 3g).

We also wish to mention that the apparent disconnection between processed Atf6 and BiP in Atase1^{-/-} mice is -at this point- unclear. A similar behavior has been reported with other cellular/tissue models. Often, non-canonical ER-stress signaling is mentioned as a possible explanation. However, we feel that in the absence of clear mechanistic evidence, it is better to just report the data and avoid any speculation.

4) *While deletion of Atase2 in APP/PS1 female mice fails to increase their life span, the percentage of male mice that survive at 10 months of age and the number of synaptophysin and Psd-95 co-localized puncta is similar between APP/PS1;Atase1^{-/-} and APP/PS1;Atase2^{-/-} mice. Therefore the conclusion that Atase1 deletion confers a more robust rescue (page 24, line 541) is only true for females and needs to be more specific in the text. Furthermore, the authors could elaborate/speculate a little more on potential mechanisms leading to male/female differences in APP/PS1;Atase2^{-/-} mice, but not APP/PS1;Atase1^{-/-} mice.*

Response:

Across all APP-based mouse models of AD, the female phenotype is always more severe. Particularly, they die faster than the males. The reason for this is unclear, although the intracellular accumulation of toxic A β aggregates and the increased propensity to spontaneous/non-inducible seizures appear to be the most common explanations (there are many reviews that discuss this issue). This also explains why most longitudinal studies focus on males alone (i.e.; APP females do not live long enough).

The overall survival curve shows a more effective rescue in the Atase1^{-/-} background. The survival curve with females shows rescue in the Atase1^{-/-} background, but not in the Atase2^{-/-} background. The late survival of males is overall similar in the two backgrounds. However, the early survival is clearly more evident in the Atase1^{-/-} background. So, overall, genetic disruption of Atase1 has a more beneficial effect than Atase2.

As we briefly mentioned above, the most likely explanation for the lethality of the APP models is the increased propensity to spontaneous/non-inducible seizures. Spontaneous/non-inducible seizures are also one of the main -ultimate- causes of mortality among AD patients. This has been linked to intracellular A β aggregates rather than the classical AD pathology itself (synaptic loss and amyloid plaques). Therefore, markers of synaptic integrity/loss or levels of amyloid plaques should not be used to exclude the sexual dimorphism of the APP models.

We also wish to point out that the severity of AD-dementia in humans does not immediately correlate with the amyloid load observed at autopsy. As such, many AD laboratories (including ours) always include the lifespan of the animals (both males and females) as a “marker” of disease severity and progression. In conclusion, we prefer to simply show all data and not speculate beyond the actual findings.

5) In discussing the data in Figs 5d and e, the authors should indicate whether the localization of the proteins and/or acetylation sites they identified are consistent with the orientation (ER lumen) of the active sites of Atase1 and 2.

Response:

The overarching scope of Figure 5d-e is to highlight the adaptive response imparted upon by changes in Atase1/2 activity. Therefore, separation of the results based on localization of the proteins would not offer any relevant information and is not important for the overall message of the paper. A more comprehensive response to this comment is provided below.

When we performed the first ever ER-specific acetylome (occupancy, not stoichiometry), we identified 549 lysine acetylation sites from 143 proteins. Of those, 60 proteins were ER-resident and 83 were ER-transiting proteins. This acetylome assessment was performed with cultured cells overexpressing AT-1 (J Biol Chem 2012; 287: 22436). When we performed the second acetylome (again, occupancy and not stoichiometry) we identified 395 lysine acetylation sites on 152 proteins, almost equally split between ER-resident and ER-transiting proteins. This acetylome assessment was performed with brain tissue from AT-1 overexpressing mice (AT-1 nTg; J Exp Med 2016; 213: 1267). In the same study, we also resolved the proteome and identified 476 proteins upregulated (only one protein was found to be downregulated). The list of upregulated proteins included ER-transiting and -resident proteins, as well as different classes of proteins that are intrinsically cytosolic. Importantly, they were almost all involved with relevant pathways (translation and post-translational modification of nascent glycoproteins; transport of nascent transiting glycoproteins across the secretory pathway; assembly of nascent transiting glycoproteins on the neuronal surface). The animals displayed expansion of the dendritic network and changes in synaptic morphology thus explaining the autism spectrum disorder-like phenotype. The interpretation of these results was as follow: *the increased availability of acetyl-CoA within the ER lumen leads to increased engagement of the secretory pathway by nascent transiting glycoproteins, to accommodate which the cell upregulates the cytosolic machinery that is required to ensure increased trafficking and peripheral assembly of the transiting glycoproteins.* This is further complicated in neurons, where assembly of synaptic structures will require (cytosolic) scaffolding and adaptor proteins, which were also found to be upregulated. In essence, the data clearly point to a significant adaptive response of the entire cell (not just the ER) triggered by changes in ER biology and imparted upon by AT-1 activity. The above interpretation is supported by several publications (Biochemical Journal 2007; 407: 383; Journal of Molecular Biology 2014; 426: 2175; Journal of Experimental Medicine 2016; 213: 1267; Aging Cell 2018; 17: e12820; Nat Commun 2019; 10: 3929; J Neurochem. 2020; 154: 404).

In conclusion, the results displayed in Figure 5d-e of the present paper reflect the adaptive response of the cell imparted upon by changes in Atase1/2 activity and are consistent with our previous findings. In other words, discriminating the adaptive response based on topology of the ATases would be very limiting and would not offer any global assessment of the cell as a whole.

6) The acetyl-CoA and CoA data reported in Fig 6f are expressed as μM or $\text{pmoles}/\mu\text{l}$. If these values refer to actual concentrations in the cytosol and ER lumen, how were the volumes of these subcellular compartments calculated?

Response:

Additional details were added in the Methods section.

7) *If the deletion of Atase1 or 2 blunts the AT1-mediated import of acetyl-CoA into the ER lumen due to the decreased production of free CoA, then accumulation of acetyl-CoA in the cytosol (even if transient, as it was not detected) should lead to similar changes in the acetylome of both KO mice. Instead, only 31 acetylpeptides were common to the two mouse strains and a large number of categories in Fig 6e were different between KO mice. The authors should expand the discussion of these results with the goal of providing an explanation. Specifically, do they expect the amount of acetyl-CoA 'accumulating' in the cytosol of the Atase1 KO and Atase2 KO mice to be different? And if this is the case, why?*

Response:

There are multiple layers of response for this question. For the sake of brevity, we will only mention two:

- 1) Not all acetyl-CoA is born equal. Since both models display partial block in the antiporter mechanism, they both should display similar adaptive response(s) to the increased availability of cytosolic acetyl-CoA. However, that was not the case. We observed differential effect with lipid droplets accumulation as well as stoichiometry of acetylation. In other words, the cell seems to differentiate between the acetyl-CoA coming from Atase1^{-/-} vs Atase2^{-/-} mice. The reason is unknown.

We recently generated SLC25A1 and SLC13A5 systemic transgenic (sTg) mice. SLC25A1 transports citrate from the mitochondria to the cytosol, while SLC13A5 transports citrate from the extracellular milieu to the cytosol. In the cytosol, citrate is then used by ACLY to generate acetyl-CoA. Overexpression of either SLC25A1 or SLC13A5 leads to increased levels of citrate in the cytosol and increased generation of acetyl-CoA. Interestingly, SLC13A5 sTg develop a segmental progeria-like phenotype that resembles AT-1 sTg mice while SLC25A1 sTg mice do not. Functional cross-talk between SLC13A5 and AT-1 is evident while functional cross-talk between SLC25A1 and AT-1 is not. Again, the cell seems to differentiate between the acetyl-CoA coming from the SLC13A5 vs. SLC25A1 pathway.

There are many instances reported in the literature where the existence of "pools" of a certain metabolite have been implicated to explain "unexpectedly" different phenotypes. This seems to be the case for the adaptive response to increased availability of cytosolic acetyl-CoA within the Atase1^{-/-}/Atase2^{-/-} and SLC13A5 sTg/SLC25A1 sTg models.

- 2) Not all acetyltransferases are born equal. As we briefly mentioned in the manuscript, different acetyltransferases display different K_m and affinity for individual peptides and lysine sites. This is quite known across all acetyltransferases (i.e.; nuclear, cytosolic, etc). In other words, different acetyltransferases are expected to respond to changes in acetyl-CoA availability differently. Therefore, it is not entirely surprising to see only 31 overlapping changes.

In conclusion, the philosophical issue raised by the Reviewer is quite complex and, although observed under different paradigms (by our group as well as others), it has no simple explanation. As such, we prefer to simply report the data and limit unnecessary speculation.

8) *In fasted mice, cytosolic acetyl-CoA is unlikely to be diverted to fatty acid synthesis, as the authors seem to suggest at page 28, lines 631-632, or in the discussion, and the accumulation of lipid droplets in the Atase2^{-/-} mice could be due to a decrease in fatty acid oxidation or*

impaired VLDL secretion. The authors should evaluate, or at least discuss, how the proteins identified in Fig 6e may be connected to lipid droplet accumulation in the liver in the fasted state (the use of fasted mice is indicated at page 28, line 630).

Response:

We apologize for the confusion with the fed vs. fasted state. The only experiment performed in the fasted state is the acetyl-CoA and CoA levels; we tried to assess these metabolites in fed mice but the variability was very high (probably due to mice eating at different times thus being at different phases of the fed-fasted continuum). All other experiments are done in the “fed” state, or more accurately, with mice that have access to food and water *ad libitum*. This has now been clarified in the Methods section.

We do recognize the lipid droplet accumulation in the Atase2 KO mice could also come from decreased fatty acid oxidation or impaired VLDL secretion. However, we have no evidence for it and it would be beyond the scope of this manuscript.

Minor

Mouse gene names should be italicized in the manuscript and figures.

Response:

Done as requested.

Reviewer #2

1. *Correction in Line 92: ‘cell degradation system’ in place of cell degrading system*

Response:

Done as requested.

2. *Correction in Line 93: replace disposing with ‘disposal’.*

Response:

Done as requested.

3. *Correction in Line 94: omit ‘across lifespan’.*

Response:

It is unclear why “across lifespan” should be omitted.

4. *Correction in Line 94: replace and with ‘whereas’.*

Response:

Done as requested.

5. *Correction in Line 102: replace but differential changes with ‘while differential changes’.*

Response:

The sentence has been modified as follows: *“Furthermore, loss of either Atase1 or Atase2 resulted in significant changes in the cellular acetylome and differential changes in acetyl-CoA metabolism”*.

6. In figure 1b: the gel for heterozygous Atase2 is showing two equal intensity bands around 1000bp, please clarify.

Response:

We do not know why this occurs but (sometimes) we do see two bands.

7. *In figure 1c: check the labeling of marker sizes*

Response:

Done as requested.

8. *Line405 should be: kidney weight, plasma creatinine and urea nitrogen,*

Response:

Done as requested.

9. *Figure 1 (Line 755): All mice under the study are male, it would be significant to observe the same events in equal number of female mice.*

Response:

This would require a considerable amount of time (for example, 7 tissue types x 3 genotypes x 3-6 mice per genotype = 63-126 RNA extractions). It is unclear what information would be added.

Figure 1c doesn't have caption.

Response:

The caption is already there.

Line 397: data should be shown for the necropsy and histologic assessment of organs.

Response:

All our mice undergo unbiased (blinded) post-mortem assessment by the UW-RARC Animal Pathology service. We receive a full necropsy and histological report, which looks just like any autopsy report (it includes mouse ID, sex, age, time of death, body weight, organ weights, carcass weight, organ examination, etc.). Following necropsy, the pathologist prepares slides for basic histology of the organs collected. The analysis is quite comprehensive and detailed. In the case of Atase1^{-/-} and Atase2^{-/-} mice, the pathologists did not detect any significant abnormalities (indeed, the animals have normal lifespan and no significant phenotype).

Including details of such a comprehensive and detailed post-mortem analysis would not offer any significant information and would be unnecessary. We included a few representative histological images here (see **Figure C** below – we show only *Atase1^{-/-}* but, if necessary we can provide *Atase2^{-/-}* as well) so that the Reviewer can appreciate them.

Figure C

10. Line 408: data should be included.

Response:

We typically outsource this analysis to the UW Analytical Laboratory. There is not much to show other than “*below limit of detection*”, which is what we said in the manuscript. A **Table** (for WT and *Atase2^{-/-}*) is included here so that the Reviewer can appreciate the results.

Animal Number	Genotype	Urine albumin (mg/L)	Urine creatinine (mg/dL)	Urine albumin:creatinine ratio (mg/g)
1	WT	6	38	15.79
2	WT	6	44	13.64
3	WT	<5*	28	Unable to calculate
4	WT	<5*	39	Unable to calculate
5	WT	<5*	44	Unable to calculate
6	WT	6	36	16.67
7	WT	7	50	14.00
8	Atase2^{-/-}	<5*	30	Unable to calculate
9	Atase2^{-/-}	<5*	28	Unable to calculate
10	Atase2^{-/-}	<5*	41	Unable to calculate
11	Atase2^{-/-}	<5*	31	Unable to calculate
12	Atase2^{-/-}	<5*	42	Unable to calculate
13	Atase2^{-/-}	<5*	38	Unable to calculate
14	Atase2^{-/-}	<5*	41	Unable to calculate

*Below limit of detection

TABLE

11. Line 440: the immunocytochemistry on *Atase1^{-/-}* knockout mice MEFs show decrease in *Fam134b* puncta (figure2b). But the literature says “downregulation of the ER-resident protein *FAM134B* prevents autophagic clearance of the ER and results in ER expansion” Abhilash I. Chiramel, Jonathan D. Dougherty, Vinod Nair, Shelly J. Robertson, Sonja M.

Best, FAM134B, the Selective Autophagy Receptor for Endoplasmic Reticulum Turnover, Inhibits Replication of Ebola Virus Strains Makona and Mayinga, The Journal of Infectious Diseases, Volume 214, Issue suppl_3, October 2016, Pages S319–S325, <https://doi.org/10.1093/infdis/jiw270>

Line 441: Also, the manuscript shows an increase in Lc3beta, which activates reticulophagy. Please explain this differential observation.

Response:

The core autophagy machinery is essentially cytosolic. However, the cell is able to activate the autophagic response in a very selective fashion to respond to the specific needs (i.e.; reticulophagy/ER-phagy to dispose of toxic protein aggregates in the ER; mitophagy to eliminate sick/damaged mitochondria, etc.). This is achieved through a series of organelle-based “receptors”, which are able to direct the core autophagy machinery to a specific location. FAM134B (as well as SEC62) appears to act as a rough ER-based “receptor” for reticulophagy.

If FAM134B (or SEC62) is a true “reticulophagy receptor”, we would expect genetic deletion of the receptor to block the induction of reticulophagy (because the autophagy machinery cannot be recruited on the rough ER). That is indeed the case (one example is the above paper mentioned by the Reviewer). However, if FAM134B (or SEC62) is a true “reticulophagy receptor”, we would also expect increased activation/progression of reticulophagy to decrease the steady-state levels of the receptor (because the receptor is being “consumed/degraded” through autophagy). That is indeed the case. Conversely, decreased activation/progression of reticulophagy would cause the opposite. That is also the case. Therefore, the levels of FAM134B seem to reflect intrinsic dynamics of reticulophagy.

Our Atase1 KO mice display reduced steady-state levels of Fam134b, which is consistent with the increased induction of reticulophagy and turnover of the ER. The increased levels LC3B-II, therefore, are in line with the above findings and argument. Importantly, our analysis involved more than just FAM134B or LC3B-II, and all the data across the different experimental paradigms (see Figure 2 and 3) are consistent.

Additional publications on FAM134B that are in line with the above are Nature 2015; 522: 354; Autophagy 2015; 11: 2377; Nature 2015; 522:359; Nature 2015; 522: 291; Nature Cell Biology 2016; 18: 1173; Nature Cell Biology 2016; 18: 1118; Autophagy 2017; 13: 322; Aging Cell 2018;17:e12820; Mol Cell. 2019; 74: 909; FEBS J 2019; 286: 2645; EMBO J 2020 ;39: e105965; EMBO J 2020;39: e102608.

12. Figure 2 (Line757) : ‘Knockout of Atase1 display activated reticulophagy’, the results shown here are with respect to enriched liver ER. Data should be included for brain samples as well.

Response:

The liver is a unique tissue. More than 98% of the cells are of the same type (hepatocytes). This is true for the entire organ (i.e.; left lobe, right lobe, etc.). The brain is not homogeneous (i.e.; neurons, astrocytes, microglia, oligodendrocytes). Furthermore, the “cellular composition” of the brain differs depending on the brain area. Hepatocytes are polarized cells that heavily depend on the secretory pathway. They also have a large cell body. Hence, they have a quite developed ER network. Not every brain cells share the same features. Although neurons are polarized and heavily depend on the secretory pathway, they have a much smaller soma and a large dendritic network. Finally, they only account for about 5% of all brain cells (in reality, this

number can go as low as 2% and as high as 10%, depending on the brain area). In conclusion, all the above, plus many other tissue-specific features, make the liver uniquely suited for ER enrichment.

13. Line 522: *'most notable in the hippocampus; we only... This should be written as 'most notable in the hippocampus; whereas we only observed...*

Response:
Done as requested.

14. Line 822: *All the results are shown only in male mice, data from female subjects should be compared.*

Response:
As discussed above, APP/PS1 females have a very short lifespan and it is very difficult to have a large number of 10-month old mice to run a solid statistical analysis. However, ThS staining of one 10-month old female per group is shown here (see **Figure D** below). The results are consistent with the males.

FIGURE D

15. Line 541: *'with knockout of Atase1 providing a more robust rescue'. But from the data represented in Figure 4, it's not really significant to assert here that Atase1-/- phenotype is a robust rescuer.*

Response:
In this portion of the manuscript, we are only highlighting the fact that the knockout of Atase1 produced a more significant rescue of the APP/PS1 phenotype than knockout of Atase2 (specifically by examining lifespan and plaque density/percent area). Indeed, the relevant paragraph within the Result section ends with the following sentence: *"Overall, our data show that knockout of either Atase1 or Atase2 in the mouse can rescue features of the APP/PS1 mouse AD-like phenotype, namely lifespan, amyloid plaque deposition, gliosis, and synapse loss, with knockout of Atase1 providing a more robust rescue."*

16. Figure 5 c should include proper labelling, the caption 'Genes' should be added on the heatmap.

Response:

Figure 5c does not show genes but proteins (they are acetylpeptides). A caption (Significant acetylpeptides) is already there.

17. Line 679: replace 'of the differing' with 'about the different'.

Response:

Done as requested.

18. Line 682: Omit the line 'Much of the consequence.....to consider'.

Response:

Done as requested.

19. Line 694: Replace 'while' with 'whereas'.

Response:

Done as requested.

20. Line 696: replace 'assume' with 'presume'.

Response:

Done as requested.

21. Line 697: add 'to' build.

Response:

Done as requested.

22. Line 707: add 'only' before 'one Atase'.

Response:

Done as requested.

Reviewer #3

The authors nicely show that the deletion of Atase1 in mouse model of Alzheimer disease reduced plaque density and neuroinflammatory signs and as consequence improve survival. There is no explanation for these results.

Response:

The reduced plaque density is likely linked to the more efficient elimination of toxic A β aggregates, although we cannot rule out an effect of the rate of A β generation (and secretion). Indeed, both APP and BACE1 are type I membrane proteins; both have a signal peptide and

insert into the ER; both are glycoproteins; both are acetylated within the ER lumen. This has already been studied (please, see *Brain* 2016; 139: 937).

Neuroinflammation is a major component of Alzheimer's diseases (as well as other neurodegenerative diseases). The mechanistic association is still a matter of speculation, although the most common (and likely) explanation is the neurodegeneration itself.

The observation that there is an activation of reticulophagy, as well as of macroautophagy in Atases KO cells is consistent with an increased ER stress due to alteration of protein acetylation. I am not convinced that this is due to a specific regulation of ATG9 by ATASE. The authors did not provided any evidence showing that ATG9a hypoacetylation is triggering activation of reticulophagy via direct interaction with FAM134B. To convincingly support this hypothesis much more data will be needed and this will be far beyond the scope of this paper.

Response:

Acetylation of ATG9A and reticulophagy. We already published that the acetylation status of ATG9A regulates the induction of reticulophagy. Importantly, "gain-of-acetylation" mutant versions of ATG9A block reticulophagy while "loss-of-acetylation" mutants have the opposite effect. Please, see *J Biol Chem* 2012; 287: 29921.

ER acetylation, ATG9A and reticulophagy. We have already published that increased ER acetylation leads to hyperacetylation of ATG9A and a block in the induction of reticulophagy while reduced ER acetylation has the opposite effect. Please, see *J Neurosci* 2014; 34: 6772; *Brain* 2016; 139: 937; *Aging Cell* 2018; 17: e12820.

ER acetylation, ATG9A and FAM134B. So far, we have published one paper showing that the acetylation status of ATG9A regulates its ability to engage FAM134B (and SEC62). Please, see *Aging Cell* 2018; 17: e12820. A follow up story where we map the area of interaction and we begin dissecting the mechanistic aspects is currently under editorial process (the manuscript is included with this submission for confidential assessment). Additional mechanistic studies are currently ongoing.

I suggest the authors to remove data 2d-f as well as the related text from the manuscript.

Response:

The data shown in Figure 2d-f are in line with our prior publications and the new manuscript currently under editorial process elsewhere (mentioned above).

Western blotting analysis in fig 3 show that samples were not loaded on the same gel. If this is the case the authors should repeat the experiments loading samples on the very same gel, otherwise the quantification is useless since different exposures will affect the results.

Response:

We apologize for the confusion. The samples were run on the same gel, although the lanes shown are not adjacent - thus a vertical line was added to the image. We have included this information in the figure legend. Please, note that we included all the uncropped gels and blots

as Supplemental Figures 6-10. The inclusion of uncropped gels is seldom necessary with the initial submission.

As for quantification, if data from multiple gels are combined (which does happen occasionally), they are first normalized both to the loading control protein (e.g. actin) then to their respective control (e.g. WT) to generate values as fold changes. Then these fold change values are combined with other fold change values from other gels. This is an accepted way to perform Western blot expression analysis.

Figure 5: Fig c, d, e, and 6d,e: I think these two figures need to be reorganized, the text is too small. I also suggest the authors to move these figures and related text in the first part of the manuscript, after the description of the atase mouse phenotype. Indeed these proteomic analysis represent the vivo investigation of Atase protein functions.

Response:

These type of plots are standard and provide a clear sense of the spectrum of changes for the general reader. More interested readers (who might be interested in the actual names listed in the graphs) will likely zoom onto the electric version of the paper.

We prefer to maintain the current order of the figures. Figure 1 provides a general introduction of the mice; Figures 2-4 describe the role of the two Atases with reticulophagy and proteostasis; Figures 5-6 describe the general adaptive cellular response to the loss of one Atase.

Reviewers' comments:

Reviewer #1 (Remarks to the Author):

The authors have satisfactorily addressed some, but not all the previous concerns. The remaining issues are indicated below.

1) While the language in lines 132-133 is indeed balanced, as the authors point out in their response to point 1) of the previous review, the title of the section (line 92) is not. It is recommended that the authors change it to 'Knockout of Atase2 resulted in a compensatory increase in Atase1 expression in multiple organs'

2) The data reported in Figure B of the response strongly support the authors' arguments for using multiple systems to answer specific scientific questions and should be included as supporting information, with the appropriate mention in the main manuscript

3) The way the authors address point 5) of the review, which was asking whether the localization of the proteins or acetylation sites are consistent with the orientation of the active sites of Atase1 and 2, is by saying that the question is irrelevant for the point they were trying to make. While that is their opinion, it is not mine or, likely, that of other readers who may wonder the same thing while reading this manuscript. Since they seem to have a reasonable explanation, I suggest they now include the few explanatory sentences contained in the last, non-dismissive paragraph of their response (i.e that changes in acetylation go beyond what would be expected based on the topology of the acetyltransferases) in the discussion of their data

Reviewer #3 (Remarks to the Author):

The authors answered to my requests without performing new experiments to strenght the conclusions of their work. They did not take into consideration my concerns related to the immunoprecipitation experiments, that lack negative controls and appropriate inputs quantification. Instead of repeating the experiments they have deleted my original request from their point-by-point response letter.

In conclusion, my experimental concerns still stand.

POINT-BY-POINT RESPONSE

We wish to thank the Editor and the Reviewers for their continued commitment to the quality of our manuscript. A comprehensive point-by-point response can be found below.

Reviewer #1

The authors have satisfactorily addressed some, but not all the previous concerns. The remaining issues are indicated below.

1) While the language in lines 132-133 is indeed balanced, as the authors point out in their response to point 1) of the previous review, the title of the section (line 92) is not. It is recommended that the authors change it to 'Knockout of Atase2 resulted in a compensatory increase in Atase1 expression in multiple organs'

Response:

Done as requested.

2) The data reported in Figure B of the response strongly support the authors' arguments for using multiple systems to answer specific scientific questions and should be included as supporting information, with the appropriate mention in the main manuscript

Response:

Done as requested.

3) The way the authors address point 5) of the review, which was asking whether the localization of the proteins or acetylation sites are consistent with the orientation of the active sites of Atase1 and 2, is by saying that the question is irrelevant for the point they were trying to make. While that is their opinion, it is not mine or, likely, that of other readers who may wonder the same thing while reading this manuscript. Since they seem to have a reasonable explanation, I suggest they now include the few explanatory sentences contained in the last, non-dismissive paragraph of their response (i.e that changes in acetylation go beyond what would be expected based on the topology of the acetyltransferases) in the discussion of their data

Response:

Done as requested.

Reviewer #3

The authors answered to my requests without performing new experiments to strenght the conclusions of their work. They did not take into consideration my concerns related to the immunoprecipitation experiments, that lack negative controls and appropriate inputs quantification. Instead of repeating the experiments they have deleted my original request from their point-by-point response letter.

In conclusion, my experimental concerns still stand.

Response:

We apologize for omitting a response to the comment on the negative controls for the IP of Fig 2d-f. This was just an honest mistake and was not done on purpose. Indeed, we responded to all the other questions raised by the Reviewer and added Figure C, Figure D and a Table to address the Reviewer's comments. We also included a related (unpublished) manuscript to provide additional validation to the Atg9a-Fam134b and Atg9a-Sec62 interaction.

The original comment that we forgot to address was: “*In addition, immunoprecipitations experiments did not show negative controls and inputs are not always showing same amount of proteins. I suggest the authors to remove data 2d-f as well as the related text from the manuscript.*”.

Input

While displayed separately, the input controls are satisfied by the data shown in *Figure 2a* where we performed Western blotting for Atg9a, Fam134b, and Sec62 in our enriched liver ER; these samples were the same as used for the immunoprecipitation studies in *Figure 2d-f*. In other words, these samples represent the input material.

Note that for the *Atase1^{-/-}* samples, the expression of Atg9a was reduced compared to WT; this was observed again in our immunoprecipitation studies shown in *Figure 2d-e* (bottom blot images). Despite this decrease (as well as a significant decrease in Fam134b expression shown in *Figure 2a*), we were still able to pull down significantly more Fam134b and roughly the same amount of Sec62 compared to WT (*Figure 2e*). These data are very similar to those already published in *Aging Cell 2018 17:e12820*.

For the *Atase2^{-/-}* samples, the expression level of Atg9a was variable and not significantly different from WT (see *Figure 2a*); the immunoprecipitation experiments again reflect this (more Atg9a was pulled down compared to WT in *Figure 2e* and less in *2f*). These data are also very similar to those already published in *Aging Cell 2018 17:e12820*.

Negative controls

The figure below shows co-IP of Atg9a-Fam134b and Atg9a-Sec62 with a negative control. These IP were done with enriched ER (mouse liver). NOTE: Atg9a, Fam134b and Sec62 represent the endogenous proteins.

Once again, we wish to stress that the related (unpublished) manuscript, which we included with our previous submission, reports:

- (i) Mapping of the area of interaction between ATG9A and FAM14B as well as ATG9A-SEC62, with positive and negative controls.
- (ii) Structural information on the modality of interaction.
- (iii) *In vivo* evidence of interaction at the ER.
- (iv) Biological effect of mutations that modified the interaction.

We also wish to highlight a few specific panels from the related manuscript:

ATG9A-FAM134B

- (a) Fig. 1F shows interaction of ATG9A and FAM134B on the ER membrane by SIM microscopy.
- (b) Fig. 2A shows that the *in vivo* interaction is disrupted when a specific domain of FAM134B is deleted. Fig. 2A includes a fundamental control – when a domain that is not required for interaction is deleted, there is no disruption on the ATG9A-FAM134B complex.
- (c) Fig. 2 points to specific structural requirements for the ATG9A-FAM134B interaction.

ATG9A-SEC62

- (a) Fig. 3C shows interaction of ATG9A and SEC62 on the ER membrane by SIM microscopy.
- (b) Fig. 3D shows that the *in vivo* interaction is disrupted when a specific domain of SEC62 is deleted.
- (c) Fig. 3 points to specific structural requirements for the ATG9A-SEC62 interaction.

In other words, the above -currently unpublished/under editorial process- data clearly validate the argument that ATG9A and FAM134B as well as ATG9A and SEC62 interact *in vivo*, and strongly support the immunoprecipitation experiments shown in Fig. 2d-f of the present manuscript.

REVIEWERS' COMMENTS:

Reviewer #1 (Remarks to the Author):

All concerns have been addressed.